# Targeting the Endocannabinoid CB1 Receptor to Treat Body Weight Disorders: A Preclinical and Clinical Review of the Therapeutic Potential of Past and Present CB1 Drugs

**DOI:** 10.3390/biom10060855

**Published:** 2020-06-04

**Authors:** Thomas Murphy, Bernard Le Foll

**Affiliations:** 1Translational Addiction Research Laboratory, Centre for Addiction and Mental Health, University of Toronto, 33 Russell Street, Toronto, ON M5S 2S1, Canada; thomas.murphy@camh.ca; 2Department of Pharmacology and Toxicology, University of Toronto, Toronto, ON M5S 1A8, Canada; 3Acute Care Program, Centre for Addiction and Mental Health, Toronto, ON M6J 1H4, Canada; 4Campbell Family Mental Health Research Institute, Centre for Addiction and Mental Health, Toronto, ON M5S 2S1, Canada; 5Department of Family and Community Medicine, University of Toronto, Toronto, ON M5G 1V7, Canada; 6Department of Psychiatry, Division of Brain and Therapeutics, University of Toronto, Toronto, ON M5T 1R8, Canada; 7Institute of Medical Sciences, University of Toronto, Toronto, ON M5S 1A8, Canada

**Keywords:** obesity, cannabinoid, cannabis, weight loss, BMI, preclinical, clinical

## Abstract

Obesity rates are increasing worldwide and there is a need for novel therapeutic treatment options. The endocannabinoid system has been linked to homeostatic processes, including metabolism, food intake, and the regulation of body weight. Rimonabant, an inverse agonist for the cannabinoid CB1 receptor, was effective at producing weight loss in obese subjects. However, due to adverse psychiatric side effects, rimonabant was removed from the market. More recently, we reported an inverse relationship between cannabis use and BMI, which has now been duplicated by several groups. As those results may appear contradictory, we review here preclinical and clinical studies that have studied the impact on body weight of various cannabinoid CB1 drugs. Notably, we will review the impact of CB1 inverse agonists, agonists, partial agonists, and neutral antagonists. Those findings clearly point out the cannabinoid CB1 as a potential effective target for the treatment of obesity. Recent preclinical studies suggest that ligands targeting the CB1 may retain the therapeutic potential of rimonabant without the negative side effect profile. Such approaches should be tested in clinical trials for validation.

## 1. Introduction

### 1.1. The Growing Concern of Obesity

Obesity is a serious and growing public health issue worldwide. One of the most common diagnostic measures of weight status is Body Mass Index (BMI), calculated by dividing an individual’s body mass (kg) by the square of their height (m^2^). The World Health Organization (WHO) classifies individuals with a BMI greater than or equal to 25 kg/m^2^ as overweight, and those exceeding 30 kg/m^2^ as obese. According to the WHO, obesity rates have tripled since 1975, and in 2016, 39% of the world’s adult population (>1.9 billion) were overweight and 13% (~650 million) were obese [1]. In 2014, global obesity rates in men and women were 10.8% and 14.9%, respectively, and are estimated to increase to 18% and 21%, respectively by 2025 [2]. It is well known that obesity plays an integral role in the development of many diseases, including diabetes mellitus and cardiovascular disease. The aggregate medical costs associated with the treatment of these diseases, while already severe and taxing on society, will only rise with an increasing obesity rate [3]. Thus, concerted efforts towards the development of novel therapeutic strategies for the treatment of obesity are vital.

### 1.2. The Endocannabinoid System and Body Weight

The endocannabinoid system (ECS) is a biological system that has been implicated in various homeostatic processes within the body, including the regulation of appetitive behaviour [4]. The ECS is composed of two main cannabinoid receptor subtypes, CB1 and CB2, two major endogenous lipid-based ligands, 2-arachidonoylglycerol (2-AG) and anandamide (AEA), and all of the enzymes involved in their synthesis and metabolism [5,6,7]. The CB1 receptor, while highly expressed in the brain and central nervous system, is found in other parts of the body, including the liver, skeletal muscle, pancreas, and adipose tissue [8]. CB2 receptors were classically considered to be located in the cells of the immune system, however, research in recent years has identified CB2 expression in areas such as the GI tract, peripheral nervous system, adipose tissue, and liver [9]. More recently, CB2 expression was detected in the brain, however, at much lower concentrations than CB1 [10]. Both CB1 and CB2 are G protein-coupled receptors, which experience conformational changes upon agonist binding: 2-AG is a full agonist of both CB1 and CB2, and AEA is a high-affinity partial agonist of the CB1 receptor and full agonist of vanilloid receptors [11,12].

ECS involvement in body weight regulation and metabolism extends from the central brain circuitry all the way to the peripheral organs involved in digestion and energy storage. The central nervous system (CNS) is highly involved in feeding, as the major responsibilities of the system include processing sensory information and assessing energy needs. Among its many functions, the feeling of hunger is mediated by the hypothalamus in the brain, triggered by hormone imbalances, such as elevated ghrelin and decreased leptin in circulation, and also by the binding of 2-AG and AEA to CB1 receptors of the hypothalamus [13]. In an obesogenic state, cases of endocannabinoid overactivity, particularly elevated 2-AG levels, have been documented and thus may exacerbate feeding and weight issues [14,15].

The peripheral nervous system (PNS) also plays an essential role in feeding as information is relayed from the periphery to the CNS via ghrelin and leptin, which are modulated by energy status and fat composition, respectively. Moreover, the PNS is involved in modulating metabolism and digestion as it assembles interactions from organs and systems, including the gastrointestinal (GI) tract, pancreas, and adipose tissue. Endocannabinoid binding to CB1 receptors in the GI tract promotes nutrient uptake as GI motility and vasodilation increase and inflammation and acid secretion decrease [16,17]. The pancreas plays a pivotal role in digestion as it is responsible for producing and secreting digestive enzymes into the GI tract. CB1 receptors are present in the insulin-producing β-cells of the islets of Langerhans and it is generally thought that endocannabinoid binding to the β-cell CB1 receptors blocks the action of insulin as the endocannabinoid-bound CB1 receptors form a heterodimeric complex with insulin receptors [18]. Finally, adipose tissue is composed of three different adipocyte cell types: white adipocytes are predominantly involved in fat storage; brown adipocytes are metabolically active and increase caloric expenditure through thermogenesis; beige adipocytes are transitional and able to transform into white or brown adipocytes in response to various stimuli. Brown adipocytes are mitochondria-rich and induce thermogenesis by uncoupling oxidative phosphorylation from ATP production using mitochondrial uncoupling protein-1 (UCP1) [19]. Interestingly, endocannabinoid-mediated CB1 activation in white adipocytes inhibits thermogenesis and, in turn, the pharmacological blockade or genetic ablation of these CB1 receptors can cause trans-differentiation into beige and brown adipocytes [16,20]. This process is referred to as “browning”.

Tetrahydrocannabinol (THC), the primary psychoactive component of cannabis, is a partial agonist of the CB1 receptor. Appetite stimulation following acute cannabis consumption is well documented and has prompted the clinical usage of cannabis to treat symptoms of suppressed feeding, such as in HIV/AIDS-related cachexia [21,22,23]. Preclinical research studies have attempted to explain this phenomenon by displaying that acute energy intake is elevated by the CB1 receptor agonists, and is inhibited by the CB1 receptor inverse agonists [24,25,26]. Therefore, it is likely that THC is responsible for the elevated feeding patterns that are characteristic of cannabis consumption. Interestingly, by analyzing the relationship between cannabis use and body weight, we have revealed a more complex relationship [27]. We explored this issue using the National Epidemiological Survey on Alcohol and Related Conditions (NESARC) and the National Comorbidity Survey-Replication (NCS-R) together, allowing for a study in excess of 50,000 US adults. Within these data sets, the incidence of obesity as a function of cannabis use led to the finding that self-reported frequent cannabis users (>3 times/week) had significantly lower obesity rates (14%/17%) than individuals that had not used cannabis in the last 12 months (22%/25%) (first value in each set is for NESARC and the second is for NCS-R) [27]. Given the size of the datasets and that significance was retained following corrections for age, sex, and tobacco consumption, these findings were robust. Other groups have since replicated these findings and even discovered lower waist circumferences in current cannabis smokers compared to former or never users [28,29,30]. Following this finding, we proposed that THC could be used to promote a reduction in body weight, a hypothesis that still remains to be tested [31].

Following the discovery of its influence over the regulation of feeding and weight, the endocannabinoid system, specifically CB1, was investigated as a potential target for anti-obesity pharmacological intervention. Rimonabant (SR141716A) is the most well-known and thoroughly studied of the family of CB1 receptor inverse agonists that were developed for obesity management. While preclinical and clinical research efforts displayed remarkable promise in rimonabant’s ability to promote weight loss in obese individuals, it was ultimately removed from the market due to the high incidence of adverse psychiatric side effects, including elevated levels of anxiety, depression, and suicidality [32]. An in-depth review of research on rimonabant and its effect on body weight will be found in the body of this review. The promise from rimonabant has led to the development of additional inverse agonists, and other compounds such as “peripherally restricted” inverse agonists, and neutral antagonists to interact with the CB1 receptor. This review article will discuss the current state of CB1 receptor-acting compounds, including cannabis, and will focus on their direct effect on body weight in preclinical and clinical research settings. Major findings from all mentioned studies are summarized in Table 1 and Table 2, outlining preclinical and clinical research studies, respectively.

## 2. The Relationship between Cannabis Use and Body Weight

### 2.1. Preclinical Research on the Effects of Cannabis Use on Body Weight

Few preclinical investigations exist that examine the direct impact of crude cannabis consumption on body weight because the investigators conducting preclinical studies of this nature have historically opted for the administration of cannabis extracts, such as purified THC, or synthetic cannabinoids. One preclinical research study, conducted by Rusznák et al., made a point of studying crude cannabis and its effect on a preclinical model of chronic stress [33]. Mice were exposed to daily chronic stress over the course of eight weeks and, during chronic stress sessions, crude cannabis was burned, exposing the mice to full-body cannabis smoke. Interestingly, the mice exposed to cannabis smoke but no stress experienced significantly lower body weights than the control group that experienced neither cannabis smoke nor stress. Furthermore, the mice exposed to stress but no cannabis smoke exhibited the lowest weights; however, cannabis exposure to the stressed cohort of mice appeared to alleviate some of the weight loss attributed to stress [33]. The results from this study suggest that cannabis had a regulatory effect on body weight.

### 2.2. Clinical Research on the Effects of Cannabis Use on Body Weight

Over the last few decades, numerous clinical research projects have investigated the relationship between cannabis use and body weight in various populations. Likely due to the illegal status of cannabis throughout most of the world, few projects have directly administered cannabis to participants in a clinical setting. Three such studies were found—two reported an increase in participant body weight as a function of cannabis consumption, and one reported that cannabis consumption significantly increased caloric intake but caused no changes in body weight [21,115,116]. Experiments that analyze participant BMI values as a function of self-reported cannabis usage are more common. As previously mentioned, robust evidence for a negative correlation between the frequency of cannabis use and BMI was discovered using the NESARC and NCS-R datasets [27]. Subsequently, other studies were conducted, many of which yielded the same relationship [117,118]. One study analyzed the National Health and Nutrition Examination Survey (NHANES) (n = 4657), reporting cannabis users as having lower fasting insulin and glucose levels, lower BMIs, and smaller waist circumferences [119]. Evidence from a prospective study in an Australian population reinforces the notion that the risk of obesity decreases with more frequent cannabis use [28]. Conversely, an increasing trajectory in adolescent cannabis use has been associated with an increased likelihood of obesity in early adulthood [120]. Interestingly, when comparing BMI-matched cannabis smokers and non-smokers, cannabis smokers have greater abdominal visceral fat accumulation, despite the fact total body fat is comparable [121].

In more recent years, a plethora of research has been published that has furthered this body of work. Experiments with the familiar self-report cannabis usage and compare-to-BMI format have been conducted, however, study populations have been separated by specific criteria, such as race or age. One study surveyed an older, economically challenged subset of the African American population of Los Angeles to uncover the health determinants of cannabis use in this population. In total, 340 participants, all aged 55 years or older, attended an in-person interview. Within this population, only 9.1% reported current cannabis use, however, current use was negatively associated with obesity [122]. Another cross-sectional study of African American participants discovered an insignificant trend towards lower BMI in current cannabis users, yet found a significant difference in waist circumference between current, former, and never users (32.9 ± 0.66 in, 35.9 ± 0.88 in, and 33.4 ± 0.74 in, respectively) [30]. The authors admit that they are uncertain whether the decrease in waist circumference is attributed to a decrease in visceral or subcutaneous fat. The Inuit people are an indigenous group that inhabit the arctic regions of North America and Greenland. Analysis of cross-sectional data from 786 Inuit survey respondents found that 57.4% of the population had used cannabis in the last 12 months and usage was associated with lower BMI and lower fat mass percentage, even though energy intake was no different [123]. With an odds ratio for obesity in past-year cannabis users of 0.56, these findings are intriguing, considering a staggering 29.3% of the Inuit population are obese and 59.2% are overweight. One study, which recruited and studied adolescent cannabis users (n = 238) suggests a significant, positive correlation between cannabis use and BMI (*p* < 0.05) [124].

Other studies have adopted a longitudinal approach to assess the effect of cannabis on body weight over a longer period. For example, we discussed the analysis of Wave 1 of the NESARC, and the negative correlation between cannabis use and BMI [27]. This survey, conducted between 2001–2002, is referred to as Wave 1, as a second round of the survey was administered from 2004–2005 to eligible Wave 1 participants. Analysis of the 33,000 participants involved in Wave 2 discovered all of the subgroups exhibited increasing BMI over this time period, but persistent cannabis users displayed the greatest attenuation in BMI gain (−0.45 kg/m^2^) compared to the quitting (−0.36 kg/m^2^) and initiating (−0.24 kg/m^2^) groups, using never-users as a reference [125]. A longitudinal assessment of men (n = 253), prospectively assessed from ages 7–32, found a negative correlation between years of daily cannabis smoking and BMI, with men using cannabis daily in excess of 10 years having the lowest BMI [126]. This finding was independent of childhood BMI, diet, and level of physical activity. The Coronary Artery Risk Development in Young Adults (CARDIA) began in 1985 and is an American longitudinal observational study, assessing the progression of cardiovascular disease risk factors. One group analyzed results from the 25-year follow-up (n = 2902) and found that years of cannabis use was inversely associated with BMI. Additionally, participants with the most years of cannabis usage (>5 years) had the lowest volumes of abdominal and subcutaneous fat, but these findings did not withstand corrections for age, sex, race, and education [127]. A similar study utilizing data from the NHANES revealed that lower BMI and waist circumference was characteristic of current cannabis users [128]. A 15-month longitudinal study of young Swiss men found that a higher BMI increased the chances of hazardous cannabis use, defined as usage two or more times in a week [129]. While this is an interesting finding, it provides little information about the BMI trajectory. Longitudinal studies focused on adolescents have found no significant association between cannabis usage in adolescence and effect on BMI moving into midlife [130].

Within this field of research, deep analyses of at-risk populations, including mentally ill and chronically ill subjects have become increasingly popular in recent years. One Spanish group conducted a three-year longitudinal study on the effect of cannabis on first-episode non-affective psychosis patients. All subjects were treated with oral antipsychotic medication upon enrolment. At baseline, cannabis-using psychoactive patients had lower BMIs than non-users (22.34 ± 3.07 vs. 23.69 ± 4.12 kg/m^2^), and this was retained in continued users at the three-year follow-up (25.06 ± 5.05 vs. 27.32 ± 4.86 kg/m^2^) [131]. Indeed, those patients that discontinued cannabis use in the three-year period exhibited the largest increase in BMI of all groups [132]. These findings are interesting, considering the weight gain characteristic of antipsychotic use. A longitudinal study of the first 12 months of treatment with antipsychotic medication replicated the finding that cannabis use curtails the expected increase in BMI [133]. From an investigation of a Dutch population (n = 3169) with severe mental illness, cannabis users had the lowest initial BMI values, and those that discontinued cannabis use had the largest increase in BMI (with a mean of 14 months between assessments) [134]. However, during this study, cannabis users had the most severe psychotic symptoms. Analysis of Parkinson’s Disease and Multiple Sclerosis patients found that cannabis users in each disease category had significantly lower BMIs than non-users (*p* < 0.035), albeit this was a self-reported web-based investigation [135].

The effect that cannabis use has on insulin resistance and diabetes has also been a topic of recent interest. Using the NHANES dataset, one group examined the relationship between cannabis use and insulin resistance in individuals stratified by BMI. Their main outcome measure for insulin resistance was fasting insulin (FINS). Of the 129,509 participants, 50.6% of current cannabis smokers were categorically obese, compared to 68.1% of never users and 79.5% of users that had abstained for ≥10 years (*p* < 0.001). The median plasma FINS concentration of cannabis non-users was 9.83 μU/mL versus 7.70 μU/mL in current cannabis users, and concentrations were higher in the long term abstained cannabis users than those who had abstained for less than a year [136]. Data from this study suggest that cannabis usage plays a protective role in the development of diabetes in obese adults by retaining insulin sensitivity. A similar study analyzed a Swedish population (n = 17,967) over the course of eight years, yielding 58.5% of non-users with a BMI ≤ 24.9 kg/m^2^, compared to 77% of participants that had used cannabis in the past year [137]. From this analysis, current or past cannabis use was inversely associated with type 2 diabetes (OR = 0.68), but this effect was lost post-correction for age (OR = 0.94).

When interpreting and comparing findings from observational studies of cannabis use, it is important to consider potential confounding factors that can be attributed to this experimental approach. Firstly, tobacco consumption is known to be negatively correlated with bodyweight, and while tobacco consumption cannot be controlled in an observational study, it must be considered. During analysis of the NESARC and NCS-R survey data, supplementary analysis was conducted to take tobacco consumption into account and the results did not affect the significance of the negative correlation between frequency of cannabis use and BMI [27]. Another limitation of observational studies is a frequent lack of consideration for different methods of consumption and variable potency. Between North America and Europe, there are crucial differences in consumption patterns and cannabis purity that need to be considered. For instance, vaping cannabis oil and the consumption of edible cannabis products have become increasingly popular worldwide, however, they are more prominent in North America than in Europe [163]. Additionally, European cannabis users commonly dilute cannabis cigarettes by mixing tobacco with the crude cannabis, a practice that is less common in North America [164]. Finally, while the THC concentrations of cannabis are on the rise in both North America and Europe, there is evidence to suggest that North American cannabis has higher concentrations of THC [165]. Future studies will need to take tobacco use, the method of cannabis consumption, and cannabis strength into account.

The study of crude cannabis is helpful due to its complex nature and composition of dozens of various cannabinoids. It is possible that the combination of the cannabinoids is responsible for its effects on body weight, however, comparison between crude cannabis and isolated molecules will help to elucidate the paramount components responsible for modulating body weight.

## 3. Exploring the Direct Effects of Cannabinoid Drugs on Body Weight

### 3.1. CB1 Inverse Agonists

Inverse agonists are compounds that elicit an opposite response to that of an agonist when binding the same receptor. CB1 inverse agonism became a popular field of research following the discovery that the blockage of the CB1 receptor reduces feeding and induces more favourable obesity outcomes [166]. Rimonabant (SR141716A) was the first clinically researched CB1 receptor inverse agonist, remaining the most widely studied drug of its type. As mentioned, rimonabant was found to successfully curb food intake and obesity in preclinical models and was ultimately graduated to clinical research settings on a large scale. Unfortunately, the clinical usage of rimonabant caused adverse psychiatric side effects, especially in patients with a history of depression [167]. Soon after, rimonabant was removed from the market in over 60 countries, including the European Union. Following the assessment of the adverse side effects of rimonabant, interest shifted to alternative inverse agonists, namely peripherally restricted inverse agonists. Here we will discuss the most widely studied inverse agonists, including rimonabant, and discuss promising novel compounds from the past few years.

#### 3.1.1. Rimonabant: Preclinical Research

Early investigations into rimonabant (K_i_ = 1.98 ± 0.13 nM) displayed its efficacy in reducing feeding and modulating energy expenditure, both of which induced weight loss in preclinical models of both lean and obese animals [25,34,35,36]. What is more, rimonabant has been shown to prevent weight gain associated with antidepressant use, decrease compulsive eating, and even enhance thermogenesis from brown adipose tissue [37,38,39,40]. Given its potential in reducing rates of obesity, more recent preclinical investigations have further expanded this work.

To further understand the ECS and rimonabant, a preclinical investigation attempted to determine whether baseline endogenous AEA levels and body weight had any influence over the weight-loss efficacy of rimonabant. Diet-induced obese (DIO) and diet-resistant (DR) rats were treated with rimonabant for 14 days. At both baseline and following treatment, the DIO rats had significantly higher AEA levels (3.5 ± 0.1 nM and 2.5 ± 0.1 nM) compared to the DR rats (2.8 ± 0.1 nM and 1.9 ± 0.1 nM), and greater baseline AEA levels correlated with larger decreases in body weight (*p* < 0.0001, r^2^ = 0.50). Baseline body weight was correlated with the therapeutic effect of rimonabant, as none of the DR rats exhibited a body weight decrease in excess of 6%, while most DIO rats exceeded this threshold, some even reaching a 12% decrease in baseline body weight [41]. The efficacy of rimonabant coadministration to attenuate the side effects of antipsychotic medications was tested on a rat model administered olanzapine. Olanzapine is an atypical antipsychotic medication used to treat schizophrenia and bipolar disorder, and characteristically induces weight gain. All rats exhibited weight gain and food intake increase during the 15 days of olanzapine treatment, however, rimonabant coadministration (10 mg/kg) was started on day 15, and by day 35, food intake had decreased and body weight was restored to vehicle group levels (*p* < 0.0001) [42].

Lipolysis is the process by which fat tissue dissipates, whereby the major components of fat tissue, triglycerides, are broken down into glycerol and fatty acids (FAs). Considering the integral role this process plays in fat accumulation and obesity, the direct effect of rimonabant on lipolysis was investigated to elucidate the drug’s underlying mechanisms and efficacy. Cultured rat adipocytes were treated with rimonabant under various conditions and the release of glycerol and fatty acids was monitored. Adipocyte treatment with increasing concentrations of rimonabant (0–10 µM) increased plasma glycerol and fatty acids concentrations from 0.05 to 0.20 mM and 0.06 to 0.45 mM, respectively [43]. Displaying the concentration-dependent increase in lipolysis from rimonabant is an interesting perspective of study as this is one of the major interactions causing clinically meaningful weight loss. An investigation of a rat model of severely uncontrolled diabetes found that rimonabant administration induced no significant weight loss, yet rimonabant induced liver amelioration by decreasing hepatic fat accumulation, ALT and AST liver enzyme levels, and cell death [44].

Recent years have seen further scientific efforts to study the effect of rimonabant on preclinical models of obesity. Consistent with previous results, the DIO mice treated with rimonabant experienced attenuation in obesity, compared to the vehicle-treated DIO mice. The first nine days of rimonabant treatment presented a nearly 60% decrease in energy intake. Energy intake soon increased to similar levels with the vehicle mice, yet the rimonabant-treated mice had significantly decreased body mass [45]. Another study using DIO mice of the same variety found that the rimonabant-treated mice were 12.7 g (17%) lighter than the vehicle-treated mice after 30 days of treatment, and 75% of the decrease was due to the loss of fat tissue [46]. An investigation into the effect of rimonabant on lipid metabolism found that the DIO mice consuming a high-fat diet while treated with rimonabant weighed an average of 30.3 ± 1.2 g while those on the same diet, treated with vehicle weighed an average of 36.5 ± 0.8 g (*p* < 0.05). More interestingly, the liver masses of the rimonabant-treated DIO mice fed a high-fat diet were significantly lower than those of the DIO vehicle-treated mice, and were even similar to those of lean mice fed a standard diet (*p* < 0.05) [47].

Chronic, low-grade inflammation is characteristic of obesity, and while rimonabant has been shown to induce weight loss, its effect on the associated inflammation required further investigation. Reductions in weight and fat mass after four weeks were again significant in rimonabant-treated DIO mice fed a high-fat diet, yet lean tissue mass was retained and not significantly different to the vehicle treated-mice (20.08 ± 0.29 g vs. 22.9 ± 0.11 g). Furthermore, rimonabant proved to decrease inflammation by downregulating several microRNAs in adipose tissue macrophages, inducing an anti-inflammatory cascade [48].

Finally, it is thought that the psychiatric side effects of rimonabant stem from its interaction with the central nervous system. For this reason, peripherally acting CB1 inverse agonists have been of interest. One study assessed the peripheral actions of rimonabant to elucidate its interactions in order to drive future compound development. Mice fed a high-fat diet gained weight, and this weight gain was rescued by rimonabant treatment (*p* < 0.05). The investigators then focused on the skeletal muscle cells of rimonabant-treated mice, and found high voltage-activated Ca^2+^ channels (HVACCs), specifically, Ca_v_1.1 was downregulated in HFD mice, and rimonabant increased Ca_v_1.1 expression in skeletal muscle cells, possibly acting as one of the obesity protective effects of the drug [49].

#### 3.1.2. Rimonabant: Clinical Research

Clinical research of rimonabant has been extensive as the drug was recognized as a promising aid in obesity treatment. Considering the fact that rimonabant was removed from the market in 2008, many of these studies are less recent, however, we will discuss the findings from the main studies that were conducted. Some of the first large-scale studies were the Rimonabant in Obesity (RIO) studies, which were randomized, double-blind, placebo-controlled, multicentre studies, occurring in both Europe and North America. Between 2001 and 2002, the RIO Europe trial enrolled and randomized 1507 obese men and women (BMI ≥ 30 kg/m^2^) and those overweight (>27 kg/m^2^) with at least one comorbidity to receive a daily administration of 20 mg rimonabant, 5 mg rimonabant, or a placebo. All participants were prescribed a diet of 600 kcal/day below their basal metabolic rate. In total, 920 study participants completed the first year of pharmacotherapy, and a significant, dose-dependent increase in weight-loss from rimonabant treatment was uncovered. The 20 mg/day group had a mean weight change of −8.6 ± 7.3 kg, while there was a −4.8 ± 6.2 kg change in the 5 mg/day group, both reaching *p* < 0.05 compared to the placebo group [138]. The proportion of subjects that lost ≥10% of baseline body weight was significantly larger in the 20 mg/day group compared to the placebo group. The RIO-Europe study was continued for a second year, with 684 completers. Neither the 20 mg/day nor 5 mg/day rimonabant groups saw significant weight loss between years 1 and 2, however they maintained the weight they had lost in the previous year (−7.2 ± 8.1 and −4.6 ± 7.6 kg, respectively) [139]. In spite of this, other cardiometabolic risk factors, such as HDL cholesterol and triglyceride levels, improved during this time. In accordance with this study, a RIO-North American trial of the same design was conducted between 2001 and 2004, enrolling 3045 subjects from the US and Canada. Similar to the European study, the subjects who were administered 20 mg of rimonabant daily for one year experienced significant weight loss over the placebo group, −6.3 ± 0.2 kg and −1.6 ± 0.2 kg, respectively (*p* < 0.001), and 25.2% of the high-dose subjects achieved weight loss ≥10% baseline bodyweight, versus 8.5% of the placebo group [140]. After the second year of pharmacotherapy, the high-dose rimonabant subjects retained their weight loss and, interestingly, those originally in the high-dose group who were switched to a placebo experienced weight gain in year 2. Throughout the RIO studies, rimonabant was generally well tolerated, however, complications became more common at higher dosages. Common adverse events included depressive symptoms, mood alterations, anxiety, and nausea, leading to a high-dose group dropout rate of 13.8% [141]. Rimonabant was even clinically determined to decrease liver fat in proportion with total body weight-loss [142]. The Comprehensive Rimonabant Evaluation Study of Cardiovascular Endpoints and Outcomes (CRESCENDO) study began in 2005 and was structured similar to the RIO studies, but enrolled a much larger cohort (n = 18,695). The study was ultimately discontinued in 2008 due to abundant adverse events, including both neuropsychiatric and serious psychiatric side effects in 32% and 2.5% of the rimonabant groups, respectively [143].

Rimonabant treatment was also studied in a population with type 2 diabetes (n = 20) and treatment over the course of six months resulted in a mean of 4% reduction in body weight (*p* < 0.001) and subjects that used insulin were able to decrease their daily dose from 116 ± 59 to 102 ± 71 units/day (*p* < 0.05) [144]. The ARPEGGIO study was a 48-week long, multicentre, double-blinded, placebo-controlled trial to determine the effect of rimonabant in type 2 diabetes patients. There were 112 completers in the rimonabant active group (20 mg/day), experiencing a mean weight change of −2.49 ± 0.31 kg versus 0.13 ± 0.26 kg of the 93 placebo group completers [145]. The RIO-Diabetes trial was a derivative of the other RIO trials, enrolling 1047 type 2 diabetic subjects and, similarly, weight loss was dose-dependent between the 5 mg and 20 mg/day rimonabant groups (−2.3 ± 4.2 kg and −5.3 ± 5.2 kg, respectively) [146]. From these promising results, there was hope that rimonabant would become an effective treatment for obesity in patients with type 2 diabetes.

#### 3.1.3. Taranabant: Preclinical Research

Around the same time rimonabant was discovered, another compound named taranabant (K_i_ = 0.13 ± 0.01 nM) was developed by Merck as part of a program to develop novel CB1 inhibitors and inverse agonists. Taranabant’s discovery stemmed from a high throughput screen (HTS) of lead compounds, eventually developing the novel acyclic amide [168]. Taranabant was preclinically studied for its effect on obesity and, similar to rimonabant, proved to be effective at promoting weight loss. Mouse model work displayed taranabant dose-dependently decreased food intake and inhibited body weight gain, as 1 mg/kg and 3 mg/kg dosages decreased overnight body weight gain by 48% and 165%, respectively (*p* < 0.01 and *p* < 0.00001). Upon further comparison between wild-type and CB1 receptor knockout mice, they found that a 3 mg/kg taranabant dose decreased overnight body weight gain by 73% (*p* < 0.00005) in wild-type mice, while no significant changes were observed in the knockouts. In DIO mice, the daily treatment of taranabant over two weeks similarly caused significant dose-dependent increases in weight loss of −3 ± 6, −6 ± 4, and −19 ± 6 g in 0.3, 1, and 3 mg/kg doses, respectively. During this same time, the vehicle-treated DIO mice gained an average of 15 ± 4 g [50]. These results have since been replicated, discovering that taranabant induces similar weight loss in both lean and DIO mice, compared to rimonabant, which was more effective at promoting weight loss in obese mice [51].

#### 3.1.4. Taranabant: Clinical Research

Considering the interest in CB1 receptor inverse agonists as a treatment for obesity, similar to rimonabant, taranabant was also extensively clinically studied. Possibly the most extensive clinical investigations of taranabant were a pair of dose-ranging, double-blinded, placebo-controlled, multicentre trials, the first of which randomized subjects to 0.5, 1, or 2 mg taranabant/day, and the second dosed 2, 4, or 6 mg taranabant/day. The low-dose study saw 693 completers and significant weight loss in all dosage groups at the end of 52 weeks: −5, −5.2, and −6.4 in the 0.5, 1, and 2 mg/day groups, respectively, compared to −1.4 in placebo group (*p* < 0.001 for all doses) [147]. The high-dose study had similar results at 52 weeks, exhibiting changes from baseline body weight of −4.1, −8.8, −10.3, and −11.5 kg in the placebo, 2, 4, and 6 mg/day groups, respectively. The 6 mg group was discontinued during the first year of treatment because the increase in efficacy over the 4 mg group was deemed too small to justify the higher incidence of adverse events. Weight loss in the 2 and 4 mg groups did not increase significantly in the second year of treatment [148]. Another phase III clinical trial took an interesting approach, having all of its subjects undergo six weeks on a low-calorie diet, and only those able to lose ≥6% of their baseline bodyweight were randomized to receive taranabant or a placebo. For those subjects randomized, pharmacotherapy was administered for 52 weeks. Weight changes following the initial six weeks of low calorie diet were significant, yet small (0, −0.5, and −1.4% for 0.5, 1, and 2 mg/day taranabant, respectively), however, the placebo group experienced some rebound in weight [149]. The main efficacy of taranabant displayed by this trial is the promotion of sustained weight loss following the low-calorie diet. It has been suggested that the weight loss efficacy of taranabant is partly due to decreased food intake and an increase in resting energy expenditure, detectable up to five hours post administration [150].

In order to assess the safety and tolerability of taranabant, clinical trials were conducted with a focus on the assessment of drug pharmacokinetics, pharmacodynamics, and safety parameters. For instance, normal weight subjects were enrolled to receive single-dose taranabant therapy in a double-blind, placebo-controlled, alternating-panel fashion of dosages between 0.5 and 600 mg. Acute administration was not associated with changes in appetite and satiety at 4 and 24 h post-dose, and no serious adverse events were reported [151]. Daily taranabant administration in healthy subjects for 14 days exhibited similarly low incidences of adverse events, however, the frequency and severity of adverse events increased with dosage, especially above 10 mg [152]. While taranabant administration in a cohort of type 2 diabetes patients proved to promote weight loss, at 52 weeks of pharmacotherapy, adverse events, including psychiatric episodes, were significant [153]. Taranabant was eventually abandoned due to the adverse events experienced from CB1 inverse agonist pharmacotherapy, which appear similar to those produced by rimonabant.

#### 3.1.5. AM 251

One compound that was discovered around the same time as the previous inverse agonists and continues to be studied to this day is AM 251. AM 251 (K_i_ = 7.49 nM) is structurally very similar to rimonabant, and was initially shown to decrease feeding and promote weight loss in preclinical animal experiments [52,53]. Furthermore, daily AM 251 administration decreased food intake and weight gain in both fasted and non-fasted animals, and drug efficacy increased with animal age and an increasing proportion of fat in the diet [54,55]. Hypophagia, decreased adiposity, and increased energy expenditure were retained in obese rats treated daily with AM 251 [56]. A combination treatment of leptin and AM 251 further reduced body weight gain, more than AM 251 in isolation, in both high-fat and free-choice fed rats [57]. Findings such as these quickly transformed AM 251 into a promising CB1 inverse agonist for the treatment of obesity.

AM 251 continues to be preclinically researched, building on past findings. Building on the findings of leptin and AM 251 coadministration, the inhibitory effects on feeding and weight gain have been reproduced. However, simultaneous 5-HT_1B_ and 5-HT_2C_ serotonin receptor antagonism was shown to eliminate the anorectic effects. This finding supports the hypothesis that leptin and AM 251 combination treatment is modulated by serotonin pathways [58]. Glucocorticoid hormones are released upon exposure to stressors, and their upregulated circulation is believed to contribute to the development of obesity and other metabolic disorders. Wild-type mice exposed to elevated glucocorticoids (cortiscosterone) quickly developed symptoms of metabolic syndrome, including increased body weight and adiposity. However, the wild-type mice exposed to elevated glucocorticoids and simultaneous daily injections of AM 251 (2 mg/kg) experienced significantly attenuated weight gain (*p* < 0.001) [59]. Other groups have experimented and tested different forms of administration, including sub-chronic intraperitoneal administration. In this context, AM 251 displayed similar dose-dependent decreases in food intake and weight gain, especially at doses of 2 and 5 mg/kg (*p* < 0.0001 compared to vehicle) [60]. The effect of CB1 inverse agonism on kidney function was tested by the administration of AM 251 in DIO rats. The DIO rats that were administered AM 251 weighed an average of 7.4% less than the vehicle-treated obese rats, post-treatment. AM 251 treatment also significantly reduced the tubular cross-sectional diameter of the kidney, compared to vehicle (*p* < 0.05), indicating that AM 251 protects from obesity-related kidney damage [61]. Furthermore, AM 251 treatment in the DIO mice was found to substantially decrease the levels of adipose tissue inflammation, in addition to the expected weight loss [62]. Clearly, AM 251 is a promising CB1 inverse agonist, but at this time, no clinical research has been conducted.

#### 3.1.6. Promising CB1 Inverse Agonists of Recent Years with a Focus on Peripherally Restricted CB1 Blockers

For many years, CB1 inverse agonists have been a research topic of interest, with the hopes of developing safe and effective pharmacotherapy options to treat obesity; however, this research was mostly stopped when rimonabant was removed from the market. Considering the fact that we have discussed the promising CB1 inverse agonists of the past, the discussion will shift to recent compounds of this type. Firstly, TM38837 (Kd = 16 nM) is a peripherally restricted CB1 inverse agonist, and compared to rimonabant, has displayed similar weight-loss efficacy, yet has lower central nervous system penetrance, potentially leading to fewer adverse side effects [63,154]. Recently, TM38837 dosages have been explored, and adverse anxiety-related side-effects only appear at doses of 100 mg/kg, 10-times higher than recommended rimonabant dosages [64]. A pair of peripherally restricted CB1 inverse agonists, AJ5012 and AJ5018, were discovered by a South Korean group through the modification of rimonabant. AJ5012 and AJ5018 were created for the purpose of having decreased brain penetrance, achieving brain/plasma concentration ratios of ~0.2 and ~0.1, respectively, compared to rimonabant, with ratios of ~1.6 and ~5.5 in the aforementioned experiments. Compared to rimonabant, AJ5012 did not decrease food intake, however, it induced approximately 60% as much weight-loss, likely due to increased energy expenditure [65]. AJ5018 reduced the food intake and bodyweight in DIO mice to a lesser extent than rimonabant, however it had similar anti-inflammatory effects [66].

JD5037 (IC_50_ = 18 nM) is a peripherally restricted CB1 inverse agonist that was synthesized as an analog of SLV-319 (Ibipinibant), a powerful CB1 inverse agonist. JD5037 displays low brain penetrance, yet high selectivity for the CB1 receptor, and has previously been shown to reduce body weight and appetite by re-establishing leptin sensitivity [67,169]. JD5037 administration to DIO mice over a 28 day period caused an approximate 20% decrease in body weight compared to vehicle [68]. A toxicity study examined JD5037 administration in rats and dogs. The mean body weight of rats in the treatment group was up to 11% lower than the control, however, there were no significant differences in weight for the dogs that were tested [69].

Other peripherally restricted CB1 inverse agonists that have been investigated include BPR0912, TXX-522, ENP11, and MRI-1867. BPR091 (IC_50_ = 8.5 nM) chronically administered to DIO mice was as effective as rimonabant at promoting weight loss, markedly elevated the levels of mRNA associated with fat oxidation and lipolysis, and induced thermogenesis, displayed by an increase in body temperature of 0.8 °C in the 10 mg/kg group [70]. TXX-522 (IC_50_ = 10.33 ± 6.08 nM), an analog of rimonabant, exhibited approximately 2% blood-brain barrier penetrance, and dose-dependently decreased body weight and fat mass in DIO mice, with levels comparable to rimonabant [71]. ENP11 is another rimonabant analog that decreases food intake in rats as early as two hours post-administration (*p* = 0.049 and *p* = 0.048; 1 and 3 mg/kg, respectively) [72]. The decreased food intake following the acute administration of ENP11 is comparable to AM251, and more pronounced than that of rimonabant. MRI-1867 is a low brain penetrating CB1 inverse agonist developed for the treatment of liver fibrosis, and DIO mice treated with MRI-1867 for 28 days experienced significant reductions in body weight and food intake, and increased energy expenditure [68,170]. Other similar compounds include Compound 6a, BMS-725519, BMS-811064, and BMS-812204 [73,171]. All of these novel compounds require further investigation.

### 3.2. CB1 Agonists

While CB1 inverse agonists have proven effective at suppressing hunger and promoting weight loss, CB1 agonists have also been synthesized and studied for their effect on feeding and weight status.

#### 3.2.1. WIN 55,212-2

Possibly the most widely studied CB1 agonist is WIN 55,212-2 (WIN). Previously, WIN (K_i_ = 62.3 nM) has been studied for its effect on feeding and weight, and by multiple accounts, high dose administration is associated with significantly decreased food intake and slowed weight gain [74,75]. The coadministration of WIN and cisplatin in rats to study WIN’s effect on cisplatin-induced GI dysmotility rendered WIN ineffective at counteracting the experienced anorexic effects, and even intensified weight loss [76]. Another field of investigation involving WIN is coadministration with glucagon-like peptide-1 (GLP-1) acting compounds such as exendin-4 (Ex-4), a GLP-1 receptor agonist that decreases food intake. WIN and Ex-4 coadministration in rats additively reduced body weight significantly more than the control or Ex-4-injected rats (*p* < 0.001 and *p* < 0.05, respectively) [77]. Although, WIN coadministration with a GLP-1 receptor antagonist, exendin (9–39), did not synergistically decrease food intake [78]. WIN was even studied for its ability to alleviate chronic mild stress, a model for stress-induced depression in rats. While WIN was revealed to prevent CMS-related memory and cognitive deficits, it had no effect on weight loss associated with the CMS model [79].

More recently, the preclinical research of WIN has been conducted and its effect on body weight solidified. Firstly, the analgesic effects of WIN were tested by spinal administration in diabetic rats to reveal whether WIN may influence neuropathic pain associated with diabetes. Following the induction of diabetes in rats, there was a significant decline in body weight (*p* < 0.001) and acute WIN treatment had no further effect on body weight [80]. Nevertheless, WIN markedly increased the pain tolerance in diabetic rats, indicated by greater latency in response to hotplate stimulation. To test the hypothesis that CB1 receptors in the small intestine contribute to feeding via the release of satiation peptides, EC activity in the gut was pharmacologically modulated using WIN. Cholecystokinins (CCK) are satiation peptides, released from the small intestinal epithelium following macronutrient arrival in the duodenum. Lean mice fed a standard diet exhibited normal release of CCK-8 following corn oil gavage (0.69 ± 0.11 ng/mL), while WIN administration prior to gavage blocked CCK-8 secretion (0.36 ± 0.04 ng/mL; *p* < 0.05), indicating that WIN interferes with the normal satiation response to feeding [81]. Another study was conducted to study cachexia in rats, and whether endocannabinoid pharmacotherapy has any therapeutic potential. Cachexia was induced in rats by an intraperitoneal injection of AH-130 Yoshida ascite hepatoma cells (cancerous liver cells), significantly decreasing body weight compared to the control (−27.3 ± 3.5 g vs. 11.2 ± 1.3 g). The cachexia index (CI%) utilizes body weight change, tumour weight, and body weight change in control animals to detect cachexia. CI% values > 10% indicate cachexia, and AH-130 injected rats in this experiment exhibited a 38.5 ± 2.1% increase in cachexia index. WIN administered prior to cachexia induction caused no difference in food intake or body weight but caused a significant reduction in cachexia index from 38.5 ± 2.1% to 25.8 ± 2.7%. Validating previous results, WIN administered rats not exposed to cachexia displayed significant decreases in body weight (*p* < 0.05) [82].

#### 3.2.2. Other CB1 Agonists

CB1 agonism is not nearly as well characterized as inverse agonism, however, in addition to WIN there are other agonistic compounds that have been studied for their effect on weight, yielding variable efficacy and weight changes. Firstly, HU-210 (K_i_ = 0.061 nM) is a synthetic cannabinoid agonist that was developed in 1988 and was studied in the early 2000s for its potential effect on obesity [172]. Rats treated daily with HU-210 exhibited dose-dependent weight loss in the first four days, reaching weights 15.9% lower than the controls, yet the rats began to gain weight from days 5–14 of treatment [83]. HU-210 induced weight loss has been replicated, and the study of HU-210 has revealed its ability to decrease maternal weight gain during pregnancy [84,85]. Another CB1 agonist compound studied for its effect on body weight is CP-55,940, an incredibly potent synthetic CB1 agonist. CP-55,940 was administered to a rat model of activity-based anorexia (ABA) to measure its effect on body weight, discovering that ABA rats exposed to vehicle experienced a seven-day weight loss of 21.11%, while a sub-chronic daily administration of 0.03 mg/kg CP-55,940 resulted in a weight loss of 17.17% and 0.06 mg/kg in a loss of 14.68%. CP-55,940 decreased weight loss in an anorexic state, while having no effect on food intake [86]. Finally, one group studied cannabidiol-2’,6’-dimethyl ether (CBDD), a dimethyl ether derivative of cannabidiol and its effect on body weight. Using ApoE-deficient mice with compromised lipid metabolism capabilities, CBDD increased body weight gain to a greater extent than vehicle-treated ApoE-deficient mice [87]. Other cannabis-related agonistic compounds such as Δ^9^-tetrahydrocannabinolic acid-A (THCA-A) have recently been studied in preclinical models and have shown promise in modulating weight; however, this is beyond the scope of this review as it is not a CB1 receptor ligand [173].

The underlying mechanisms of the seemingly paradoxical finding that CB1 agonists, similar to CB1 inverse agonists, promote weight loss and protect against obesity have not been fully elucidated. One hypothesis is that CB1 agonists act as functional antagonists in vivo, antagonizing the endogenous cannabinoids, specifically 2-AG [31]. Considering the fact that endocannabinoid overactivity and elevated peripheral 2-AG levels are characteristic of visceral obesity, this explanation seems plausible [14,15]. This hypothesis requires further investigation.

### 3.3. CB1 Partial Agonists

#### 3.3.1. Tetrahydrocannabinol (THC)

Δ^9^-tetrahydrocannabinol (THC), as previously mentioned, is an abundant cannabinoid and the primary psychoactive component of cannabis. Over the years, there has been controversy surrounding the agonistic abilities of THC (K_i_ = 27.1 nM) at the CB1 receptor, but it has been confirmed to be a CB1 partial agonist [174]. Notably, THC also binds many other receptors including PPARγ, TRPA, and TRPV receptors [175,176]. THC is highly lipophilic, causing localization in adipose tissue, and while it is well established that THC promotes food intake, there is evidence to suggest it has an anorexigenic effects [21,24,31,177,178]. Here, we will be exploring research on the effect of THC on body weight.

THC administration to anorexic ABA mice has revealed that THC increases food intake, and attenuates anorexia-related body weight loss, compared to vehicle-treated ABA mice [88,89]. Normal weight rats administered THC have been shown to experience decreased body weight, as a daily administered dose of 10 mg/kg caused significantly diminished weight after seven days of administration (*p* < 0.05). Interestingly, despite the exhibited weight loss, the adipocytes of the THC-treated rats increased in surface area to over twice the size of the vehicle-treated rats (*p* < 0.001) [90]. In a study of type 2 diabetic rats, vehicle-treated diabetic rats experienced significant weight loss, while THC appeared to rescue this weight loss, as the THC-treated diabetic rats experienced slight, insignificant weight gain [91].

In recent years, preclinical work surrounding the study of THC and its effect on body weight has been expanded. Following the onset of puberty, rats expectedly experienced marked growth and weight gain in a study following rats in the 14 days post onset. Interestingly, while all of the control, vehicle-treated, and THC-treated rats gained weight, the rats exposed to daily THC injections of 5 mg/kg exhibited slowed weight gain on every day of testing (*p* < 0.001) [92]. Research efforts have also studied the effect of THC administration in lean versus DIO animals with mice treated with vehicle for four weeks or THC doses of 2 mg/kg for three weeks which were increased to 4 mg/kg for the final week. No effect on food intake or body weight was observed in lean mice, yet the THC-treated DIO mice experienced significant reductions in body weight (*p* < 0.001), fat mass (*p* < 0.05), and energy intake (*p* < 0.05) compared to the vehicle, preventing any changes from baseline [93]. In this experiment, THC reverted DIO-specific microbiota changes, specifically a reduction in the Firmicutes/Bacteroidetes ratio, signifying one of the possible mechanisms mediating the results. Following the previous findings of CB1 inactivation and the subsequent reductive effects on body weight, a “hyper-sensitive” form of CB1 was expressed. The S426A/S430A mutation converts serines 426 and 430 to alanines, blocking the desensitization of the CB1 GPCR, and creating an animal model with increased binding and prolonged cannabinoid signaling. Acute THC injections (1 mg/kg and 3 mg/kg) similarly increased feeding in both wild-type and mutant mice, and no significant differences in body weight were observed between genotypes [94].

THC was used in a preclinical trial to study its effect on fructose-induced liver damage. Over the course of 12 weeks, male rats were given either free access to fructose, fed a normal diet for eight weeks then treated daily with 1.5 mg/kg THC for the final four weeks, administered both fructose for 12 weeks and THC for the final four weeks, or were the control. Following 12 weeks, the THC group weighed a mean of 308.16 ± 14.67 g and the control group weighed 356.18 ± 12.36 g, compared to baseline weights of 274.46 ± 12.67 and 279.54 ± 10.11 g, respectively (*p* < 0.05) [95]. THC clearly attenuated body weight increase, yet fructose administration in isolation did not significantly affect body weight and did not induce greater body weight loss when administered alongside THC. Finally, with increasing levels of adolescent cannabis use, the study of its effect on this population has become a topic of interest. To study this, adolescent rats were placed in a sealed exposure chamber and exposed to THC vapour for 30 min, twice daily for five consecutive days, followed by a two-day break, then five more consecutive days of exposure. Male rats consumed more food throughout the trial, and during the second treatment week, the THC-exposed male rats exhibited significantly lower bodyweight than the vehicle-treated rats (*p* < 0.05) and a lower weight gain trajectory, while the female rat groups did not differ in bodyweight [96].

Preclinical findings suggest that THC has a regulatory effect on body weight despite its apparent increase in feeding. Very few clinical research studies have utilized purified THC and analyzed its effect on bodyweight, as most trials of this type opted to used crude cannabis or synthetic forms of THC, including dronabinol and nabilone.

#### 3.3.2. Dronabinol

Dronabinol is a synthetic form of THC, specifically, the same (−)-trans-Δ⁹-tetrahydrocannabinol enantiomer found in crude cannabis. Due to its identical chemical structure to THC, little to no preclinical research has been conducted on dronabinol, yet it has been the drug of choice for the clinical study of THC. As previously mentioned, it is well established that dronabinol stimulates acute food intake and reduces weight loss in clinical populations with disease states characterized by reduced appetite and significant weight loss [22,23,155,156]. Additionally, dronabinol is generally well tolerated. For this reason, dronabinol is currently produced as an appetite stimulant, antiemetic, and is an approved treatment for HIV/AIDS-related cachexia as well as chemotherapy-induced nausea [179].

Recently, dronabinol has been further studied for its effect on metabolic parameters and weight status. Anorexia nervosa has become one of the most recent conditions of study, with various clinical trials endeavoring to assess the pharmacological potential of dronabinol to treat this disorder. One randomized, double blind, placebo-controlled crossover study randomized anorexic women (n = 25) to receive dronabinol–placebo—2.5 mg of dronabinol twice daily for four weeks, a four-week wash-out period, followed by four weeks of placebo dronabinol—or placebo–dronabinol, in reverse order. During dronabinol pharmacotherapy, participants experienced a mean weight gain of 1.00 ± 1.4 kg, and 0.66 ± 1.4 kg over placebo (*p* = 0.03), representing a gain of 0.17 kg per week over the placebo [157]. Further analysis of this dataset with a focus on physical activity reveals that participants increased exercise intensity during dronabinol pharmacotherapy by approximately 20% (*p* = 0.01), accounting for 68.2 ± 126.6 kcal/day excess energy expenditure over the placebo (*p* = 0.01) [180]. Interestingly, dronabinol pharmacotherapy promoted weight gain despite the increased energy expenditure. Dronabinol was generally well tolerated.

The effect of dronabinol pharmacotherapy has also been investigated to observe its effect on metabolic parameters. This double-blinded, placebo-controlled study randomized a population presenting with non-cardiac chest pain to receive daily 5 mg administrations of dronabinol or placebo for 4 weeks. Amongst the study completers, there were no significant changes in bodyweight nor any significant changes in metabolic parameters including cholesterol, triglycerides, glucose, insulin, and leptin. These findings indicate a lack of harmful metabolic side effects resulting from this appetite stimulating agent and warrant its potential use in patients with metabolic disorders [158]. Most recently, the efficacy and safety of dronabinol was tested in an inpatient clinical setting to assess its treatment of dampened appetite and bodyweight resulting from acute or chronic illness. From a cohort of 38 patients requiring appetite stimulation pharmacotherapy, five were prescribed dronabinol, with the remaining being prescribed megestrol, mirtazapine, or a combination of these orexigenic compounds. Two received megestrol and dronabinol, and one received mirtazapine and dronabinol. The mean dronabinol usage period was 228 h, prompting a mean meal intake of 38 ± 34% at drug discontinuation, compared to meal intake of 29 ± 31% at initiation, and increased feeding in 80% of subjects [159]. There were no significant changes in body weight following dronabinol treatment. Compared to megestrol and mirtazapine, dronabinol had similar efficacy and was well tolerated, with no participants reporting symptoms of nausea or vomiting.

#### 3.3.3. Nabilone

Nabilone (K_i_ = 2.2 nM) is a potent, synthetic cannabinoid analog of THC, with greater bioavailability and a longer duration of action than dronabinol, and is an approved antiemetic for cancer patients undergoing chemotherapy [181,182]. At this time, few studies have examined the effect of nabilone on body weight and they have only studied populations undergoing chemotherapy. One clinical trial specifically studied nabilone administration in patients receiving chemotherapy for head and neck carcinomas (n = 65). Compared to the placebo group, nabilone-treated patients did not experience increased quality of life, significant weight change, decreased nausea, or improved appetite during or after chemotherapy [160]. Another clinical trial recently assessed nabilone’s effect on post-operative nausea and vomiting (PONV) following elective surgery and found no difference in PONV incidence, compared to the placebo [161]. No data on bodyweight were collected.

#### 3.3.4. AM11101

Another novel CB1 agonist studied for its orexigenic effects is AM11101. This compound was synthesized from an optimization of Δ^8^-THC through the addition of oxime and polar groups at C3 of the alkyl tail of the original compound. The binding analysis of AM11101 revealed its partial agonism of the CB1 receptor through the binding of the modified alkyl tail, yielding a K_i_ value at the CB1 receptor of 0.9 nm, compared to a K_i_ of 27.1 nm in THC [183]. AM11101 was then graduated to preclinical experimentation, where acute as well as daily administrations were examined. During this experiment, rats exposed to acute doses of AM11101 treatment (0.01, 0.05, 0.1 mg/kg) experienced increased food intake one hour post-treatment (*p* < 0.05), whereas THC had no effect on food intake. Chronic AM11101 administration over seven days caused similar increases one hour post administration feeding, yet there were no significant effects on body weight [97]. AM11101 requires further preclinical testing.

### 3.4. CB1 Neutral Antagonists

The next endocannabinoid system-acting drug class of interest is the CB1 neutral antagonist class. These compounds neither increase nor decrease CB1 receptor signaling, yet their binding blocks other compounds from binding the CB1 receptor. With their neutral effect on signaling, they are an interesting topic of study for their effect on obesity.

#### 3.4.1. LH-21

LH-21 (K_i_ = 855.6 ± 296 nM) was synthesized in 2004 and was one of the first CB1 neutral antagonists studied for its effect on feeding and weight. When administered to obese rats, LH-21 dose-dependently decreased food intake and body weight gain and, compared to rimonabant, induced similar anorexigenic effects, yet reduced the side effects of anxiety and mood disorder compared to rimonabant, plausibly due to the poor blood-brain barrier permeability of LH-21 [98]. Shortly after, the weight loss efficacy of LH-21 in DIO rats was reproduced by another group [99]. One pharmacological study investigated the mechanism of action of LH-21, claiming it is in fact a low-affinity CB1 inverse agonist, but these claims require validation [100].

Recently, LH-21 was tested as a treatment to prevent the onset of type 2 diabetes. A DIO, pre-diabetic mouse model was treated daily with LH-21 (3 mg/kg) or vehicle for two weeks, causing no significant decreases in food intake or body weight compared to vehicle. However, the LH-21 treated mice had slightly lower fasting glucose levels (98 ± 7 mg/dL vs. 108 ± 4 mg/dL, *p* = 0.193) and decreased insulin secretion in response to 11 mM of glucose (*p* < 0.001) [101]. While LH-21 did not favourably modulate weight and food intake in these animals, these findings suggest that LH-21 could be used as a treatment to prevent the onset of type 2 diabetes. Furthermore, LH-21 was assessed for its effect on obesity-induced hypertension, revealing that a daily LH-21 injection (3 mg/kg) for three weeks in DIO mice significantly decreased their mean blood pressure (*p* < 0.05) and bodyweight (*p* < 0.01), even though there was only a small, yet significant, decrease in food intake (*p* < 0.05) [102]. These findings indicate the promise of LH-21 in treating the obesogenic state, but more research is required.

#### 3.4.2. AM6545

AM6545 (K_i_ = 1.7 nM) is another CB1 neutral antagonist that was synthesized following increased interest in peripherally restricted CB1 compounds with limited blood-brain barrier permeability. Similar to LH-21, AM5445 is a pharmacologically confirmed neutral antagonist with low central penetrance that dose-dependently reduces food intake and body weight in preclinical animal models [103,104]. Similar to CB1 inverse agonists, such as rimonabant, and their ability to induce the browning of adipose tissue and augment metabolism, AM6545 neutral antagonism has been shown to stimulate metabolism through this same mechanism. DIO mice treated with AM6545 for four weeks exhibited a 19% reduction in body weight (*p* < 0.001) and 23% reduction in fat mass (*p* < 0.01) compared to vehicle, and significantly increased brown adipose tissue activity [40]. Considering the fact that AM6545 induces effective weight loss and operates under a similar mechanism to rimonabant, research into its pharmacotherapy potential has been active.

As discussed earlier, WIN administration has been shown to decrease the secretion of the CCK-8 satiation peptides, indicating its interference with the satiation response. In the same experiment, AM6545 coadministration with WIN elevated CCK-8 plasma levels from 0.36 ± 0.04 ng/mL to 0.75 ± 0.14 ng/mL, rescuing the satiation response [81]. It is interesting how AM6545 has been able to decrease body weight and adiposity while causing no significant decrease in food intake. DIO mice treated for three weeks with 3 mg/kg or 10 mg/kg of AM6545 per day experienced significant weight loss (*p* < 0.05 and *p* < 0.01, respectively) and decreased adipose tissue (−11.7% and −35.3%, respectively) with no effect on food intake [105]. Finally, glucocorticoid (GCs) signaling and its association with metabolic syndrome and obesity are well established, as it is hypothesized that stress induces greater GC signaling, contributing to the obesogenic state. Mice exposed to corticosterone (CORT) in their drinking water rapidly develop metabolic syndrome, including weight gain and increased adiposity, yet the coadministration of AM6545 and CORT blocked the expected weight gain and increase in adiposity (*p* < 0.001) [59]. This indicates the importance of the peripheral ECS in obesity and how AM6545 may prove to be a potential mediator of GC-induced obesity.

#### 3.4.3. AM4113

AM4113 (K_i_ = 0.80 ± 0.44 nM) is another CB1 neutral antagonist that has proven effective to promote weight loss in preclinical models. What is interesting about AM4113 is, unlike AM6545, it is not peripherally restricted, as AM4113 was confirmed to cross the blood-brain barrier [106]. Acute administration to rats has been shown to decrease acute food intake and, in turn, daily administration has been shown to dose-dependently decrease weight gain while inducing no signs of nausea [107]. The chronic administration of AM4113 has been shown to initially decrease food intake, induce a sudden weight loss, and, as food intake returns to normal, maintain the decreased body weight [108].

Recently, a study of AM4113 administration’s effect on an animal model of nicotine dependence revealed that, compared to rimonabant, AM4113 had similar weight loss efficacy with little to no adverse side effects. Rats were administered AM4113 (1, 3, or 10 mg/kg), rimonabant (1, 3, or 10 mg/kg), or vehicle daily for 21 days, and AM4113 doses of 3 mg/kg and 10 mg/kg effectively promoted weight loss (*p* < 0.05 and *p* < 0.001, respectively) to an extent comparable to that of 10 mg/kg of rimonabant (*p* < 0.05) [109]. Unlike rimonabant, AM4113 treatment was not found to induce any symptoms of anxiety or depression, as measured by the elevated plus maze and the forced swim test. AM4113 tested as a treatment for alcoholism has revealed its ability to decrease alcohol intake in binge-like ethanol consuming mice. A daily treatment of 1 or 3 mg/kg AM4113 in these mice revealed a dose-dependent suppression of alcohol intake, whereby on the first day of treatment, the 1 mg/kg and 3 mg/kg doses of AM4113 caused reductions in ethanol consumption of 1.19 g/kg (*p* < 0.0001) and 1.81 g/kg (*p* < 0.0001), respectively [110]. These reductions in ethanol consumption persisted for the four days of testing, however, there were no significant changes in body weight, possibly due to the short treatment period. In spite of this, the experiment provides support for use of AM4113 in the treatment of alcoholism.

#### 3.4.4. THCV

Another interesting CB1 neutral antagonist is tetrahydrocannabivarin (THCV), a homologue of THC. THCV (K_i_ = 75.4 nM) has a 3-carbon propyl alkyl group instead of the 5-carbon pentyl group of THC, causing profound differences as THCV binds the CB1 receptor as a neutral antagonist [184]. Similar to other CB1 neutral antagonists discussed, THCV administration has been shown to acutely decrease food intake and body weight effectively in both fasted and non-fasted mice [54]. Another experiment aimed at studying the effect of THCV in DIO mice and genetically obese mice, found that TCHV administration had no significant effect on body weight or food intake, but increased energy expenditure by 8.2% and 13.5% at doses of 5 and 12 mg/kg, respectively [111]. THCV also increased glucose tolerance and restored insulin sensitivity, indicating its potential use as an obesity treatment. Unlike the other neutral antagonists discussed, THCV has been applied to clinical research as the effect of a single 10 mg oral dose was studied in a randomized, within-subject, double-blind experimental design. Healthy participants (n = 19) received either THCV or a placebo, underwent an MRI blood-oxygenation-level-dependent (BOLD) scan one hour post-administration, and again one week later, receiving the other drug. The results revealed a positive correlation between BMI and increased connectivity between the amygdala and precuneus in the placebo group, a correlation that was not exhibited in the THCV group [162]. The authors hypothesize that this finding may symbolize the mechanism by which THCV is able to modulate food intake. Obviously, THCV requires further clinical investigation.

#### 3.4.5. Other CB1 Neutral Antagonists

Other less studied CB1 neutral antagonists of interest are NESS06SM, SM-11, and PIMSR. NESS06SM (K_i_ = 10.25 nM) is a peripherally restricted CB1 neutral antagonist and, when administered to DIO mice fed a high-fat diet, induces comparable weight loss and reductions in caloric intake to rimonabant while avoiding the mRNA expression changes associated with anxiety and depression [112]. Moreover, much like rimonabant, NESS06SM coadministration with the atypical antipsychotic olanzapine has been shown to offset the expected weight gain [42]. SM-11 is another neutral antagonist of CB1, belonging to the same family as NESS06SM. Intraperitoneal administration daily for 10 days revealed that the highest doses of SM-11, 0.125 and 0.25 mg/kg, reduced rat food intake by 15–20% and significantly reduced body weight compared to the vehicle-treated group (*p* < 0.0001) [113]. The antagonistic ability of SM-11 was also displayed as it was able to fully antagonize the CB1 receptor agonist activity of WIN55,212-2. Finally, PIMSR (K_i_ = 17 nM) is another neutral antagonist that, when administered daily to DIO mice for 28 days, decreased weight gain and adipose tissue development [114]. While all of these novel compounds show promise, they require further research.

## 4. Conclusions

The endocannabinoid system is complex, and many underlying mechanisms are widely misunderstood. Previous works have undoubtedly linked this system to the regulation of metabolism and body weight, establishing it as an intriguing target for efforts to develop pharmacotherapeutic tools to treat obesity. The purpose of this review was to discuss the various CB1 receptor-acting compounds that have been studied for their effect on body weight. Beginning with cannabis and transitioning into the various classes of synthetic compounds, including inverse agonists, full agonists, partial agonists, and neutral antagonists of the CB1 receptor, many of these compounds require further investigation. While rimonabant ultimately proved too dangerous for human administration, this failure ignited the creation of numerous novel inverse agonists, many of which display similar efficacy with more favourable side effect profiles. CB1 agonists have largely been revealed to decrease feeding and induce weight loss, while partial agonists increase feeding, however, induce a regulatory effect on weight status whereby overweight individuals lose weight and underweight individuals gain weight. Neutral antagonists typically induce favourable weight loss and may provide effective treatment options for metabolic syndrome and the obese state.

It is clear that more research is required to further understand the mechanisms of action and uncover the potential of the aforementioned compounds in treating weight disorders. Future investigations with crude cannabis need to employ study designs that control confounding factors, such as tobacco and substance use, mode of administration, and purity and potency of cannabis product. A transition from observational studies to the controlled clinical administration of cannabis is likely the solution and this prospect is becoming more realistic with ever increasing legalization. Many novel CB1 inverse agonists, especially peripherally-restricted inverse agonists require further preclinical research and careful consideration before promotion to clinical research to avoid similar situations to those experienced with rimonabant. The effect of CB1 agonists on body weight needs further investigation to explain the underlying mechanisms of the previously reported paradoxical anti-obesity effects. While dronabinol, the synthetic formulation of THC, has been clinically studied, research efforts are lacking. Previous clinical trials with dronabinol extended for short time intervals of one month or less and administration to an obese population has not occurred. Perhaps the next step would be the daily induction of dronabinol in obese populations for a time period of up to six months, albeit at doses small enough to negate psychotropic effects. Similar to CB1 inverse agonists, compounds of the neutral antagonist class require testing to ensure safety and tolerability prior to use in clinical research.

## Figures and Tables

**Table 1 biomolecules-10-00855-t001:** Summary of Evidence: Preclinical Studies of Cannabinoid Drug Effect on Body Weight.

Study Reference	Animal Model (Species)	Cannabinoid Administered	Cannabinoid Type	Drug Administration	Population Size	Duration	Effect on Body Weight, Compared to Vehicle (If Applicable)	Other Notes
Rusznák et al., 2018 [33]	Chronic mild stress (male NMRI mice)	Cannabis	Cannabis	Whole body smoke, 30 min, twice per day	n = 36	8 weeks	Increase	
Colombo et al., 1998 [25]	Lean (male Wistar rats)	Rimonabant	Inverse Agonist	IP injection, once daily (2.5, 10 mg/kg)	n = 19	2 weeks	Decrease	
Kunz et al., 2008 [34]	Lean (male Sprague–Dawley rats) and CB1R deficient mice)	Rimonabant	Inverse Agonist	Oral micro-suspension, once daily (2 mL/kg, 4 mL/kg)	n = 20	2 weeks	Decrease	
Richey et al., 2009 [35]	Lean (mongrel dogs)	Rimonabant	Inverse Agonist	Oral, once daily (1.25 mg/kg)	n = 20	16 weeks	Decrease	
Herling et al., 2008 [36]	DIO (female Wistar rats)	Rimonabant	Inverse Agonist	Oral, once daily (10 mg/kg)	n = 16	6 weeks	Decrease	
Gobshtis et al., 2007 [37]	Antidepressant-treated (female Sabra mice)	Rimonabant	Inverse Agonist	IP injection, 5 weekly (2, 5 mg/kg)	n = 16	Acute and up to 22 weeks	Decrease	
Dore et al., 2014 [38]	High-sucrose diet (male Wistar rats)	Rimonabant	Inverse Agonist	IP injection, once daily (0.3, 1, 3 mg/kg)	n = 44	24 days	Decrease	
Bajzer et al., 2011 [39]	DIO (male C57BL/6 J mice)	Rimonabant	Inverse Agonist	IP injection, once daily (10 mg/kg)	n = 33	7 weeks	Decrease	
Boon et al., 2014 [40]	DIO (E3L.CETP male mice)	Rimonabant	Inverse Agonist	IP injection, once daily (10 mg/kg)	n = 18	4 weeks	Decrease	
		AM6545	Neutral Antagonist	IP injection, once daily (10 mg/kg)			Decrease	
Karlsson et al., 2015 [41]	DIO and diet-resistant (male Sprague–Dawley rats)	Rimonabant	Inverse Agonist	Gavage, once daily (5 mL/kg)	n = 30	2 weeks	Decrease	
Lazzari et al., 2017 [42]	Antipsychotic-treated (female Wistar rats)	Rimonabant	Inverse Agonist	Gavage, once daily (10 mg/kg)	n = 40	5 weeks	Decrease	
		NESS06SM	Neutral Antagonist	Gavage, once daily (10 mg/kg)			Decrease	
Muller et al., 2020 [43]	Cultured adipocytes (male Wistar rats)	Rimonabant	Inverse Agonist	Bolus, single administration (30 mg/kg)	unknown	Acute	Not assessed	
Chang et al., 2018 [44]	Severely uncontrolled diabetes (LETO rats)	Rimonabant	Inverse Agonist	Gavage, once daily (10 mg/kg)	n = 20	6 weeks	No change	
Mehrpouya-Bahrami, 2017 [45]	DIO (male C57BL/6 J mice)	Rimonabant	Inverse Agonist	Gavage, once daily (10 mg/kg)	n ~ 50	4 weeks	Decrease	
Zhang et al., 2012 [46]	DIO (male C57BL/6 J mice)	Rimonabant	Inverse Agonist	Gavage, once daily (10 mg/kg)	n = 23	30 days	Decrease	
Wei et al., 2018 [47]	DIO (male C57BL/6 J mice)	Rimonabant	Inverse Agonist	Gavage, once daily (10 mg/kg)	n = 70	3 weeks	Decrease	
Mehrpouya-Bahrami, 2018 [48]	DIO (male C57BL/6 J mice)	Rimonabant	Inverse Agonist	Gavage, once daily (10 mg/kg)	unknown	4 weeks	Decrease	
Chen and Hu, 2017 [49]	DIO (male C57BL/6 J mice)	Rimonabant	Inverse Agonist	Gavage, once daily (30 mg/kg)	n = 39	5 weeks	Decrease	
Fong et al., 2007 [50]	Wild-type and CB1 knockout (male C57BL/6 J mice), and DIO (male Sprague–Dawley rats)	Taranabant	Inverse Agonist	Gavage, once daily (0.3, 1, 3 mg/kg)	n = 36 mice; n = 23 rats	2 weeks	Decrease	
Martín-García et al., 2010 [51]	DIO and lean (female Wistar rats)	Taranabant	Inverse Agonist	Sublingual, once daily (3 mg/kg)	n = 48	13 weeks	Decrease	Rimonabant and taranabant were more effective in obese mice
		Rimonabant	Inverse Agonist	Sublingual, once daily (10 mg/kg)			Decrease	
Hildebrandt et al., 2003 [52]	DIO (male C57BL/6 J mice)	AM 251	Inverse Agonist	Gavage, once daily (3, 30 mg/kg)	n = 30	6 weeks	Decrease	
Chambers et al., 2004 [53]	DIO (Lewis rats)	AM 251	Inverse Agonist	IP injection, once daily (1.25, 2.5, 5 mg/kg)	n = 8	10 days	Decrease	
Riedel et al., 2009 [54]	Wild-type (male C57BL/6 J mice)	AM 251	Inverse Agonist	IP injection, once daily (10 mg/kg)	n = 16	4 days	Decrease	
		THCV	Neutral Antagonist	IP injection, once daily (3, 10, 30 mg/kg)	n = 28	2 days	Decrease	
Judge et al., 2009 [55]	Wild-type and DIO (male Fisher 344X Brown Norway rats)	AM 251	Inverse Agonist	IP injection, once daily (0.83, 2.78 mg/kg)	n = 61	6 days	Decrease	
Merroun et al., 2013 [56]	Lean and DIO (male Zucker rats)	AM 251	Inverse Agonist	IP injection, once daily (3 mg/kg)	n = 32	3 weeks	Decrease	
Wierucka-Rybak et al., 2014 [57]	DIO (male Wistar rats)	AM 251	Inverse Agonist	IP injection, once daily (1 mg/kg)	n = 34	6 days	Decrease	AM 251 and Leptin coadministration augmented weight loss
Wierucka-Rybak et al., 2016 [58]	DIO (male Wistar rats)	AM 251	Inverse Agonist	IP injection, once daily (1 mg/kg)	n = 40	6 days	Decrease	Serotonin receptor antagonism abolished anorectic effects
Bowles et al., 2014 [59]	Wild-type and CB1R knockout (C57BL/6 J mice)	AM 251	Inverse Agonist	IP injection, once daily (2 mg/kg)	n ~ 20	4 weeks	Decrease	
		AM6545	Neutral Antagonist	IP injection, once daily (10 mg/kg)			Decrease	
Merroun et al., 2015 [60]	Wild-type (male Wistar rats)	AM 251	Inverse Agonist	Sub-chronic IP injection, once daily (1, 2, 5 mg/kg)	n = 40	8 days	Decrease	
Jenkin et al., 2015 [61]	DIO (male Sprague–Dawley rats)	AM 251	Inverse Agonist	IP injection, once daily (3 mg/kg)	n = 18	6 weeks	Decrease	
Miranda et al., 2019 [62]	DIO (C57BL/6 J mice)	AM 251	Inverse Agonist	Gavage, once daily (10 mg/kg)	n = 20	4 weeks	Decrease	
Takano et al., 2014 [63]	Wild-type (cynomolgus monkeys)	TM38837	Inverse Agonist	Intravenous (0.3–4 mg/kg)	n = 3	Acute	Not assessed	
Micale et al., 2019 [64]	Wild-type (male C57BL/6 J mice)	TM38837	Inverse Agonist	Oral, once daily (10, 30, 100 mg/kg)	n = 45	10 days	Not assessed	Study of fear-promoting effects in mice
Han et al., 2019 [65]	DIO and leptin-receptor deficient (male and female C57BL/6 J mice)	AJ5012	Inverse Agonist	IP injection, once daily (20 mg/kg)	n = 20	4 weeks	Decrease	
		AJ5018	Inverse Agonist	Not assessed			Not assessed	
Han et al., 2018 [66]	DIO (C57BL/6 J mice)	AJ5018	Inverse Agonist	IP injection, once daily (10 mg/kg)	n ~ 16	4 weeks	Decrease	
Tam et al., 2012 [67]	DIO (male C57BL/6 J mice)	JD5037	Inverse Agonist	Gavage, once daily (3 mg/kg)	n = 28	4 weeks	Decrease	
Udi et al., 2020 [68]	DIO (male C57BL/6 J mice)	JD5037	Inverse Agonist	Oral, once daily (3 mg/kg)	n = 58	4 weeks	Decrease	
		MRI-1867	Inverse Agonist	Oral, once daily (3 mg/kg)			Decrease	
Kale et al., 2019 [69]	Wild-type (Sprague–Dawley rats and Beagle dogs)	JD5037	Inverse Agonist	Rats: Gavage, once daily (10, 40, 150 mg/kg); dogs: Gavage, once daily (5, 20, 75 mg/kg)	Rats: n = 140; dogs: n = 44	34 days	Decrease in rats; no change in dogs	
Hsiao et al., 2015 [70]	DIO (male C57BL/6 J mice)	BPR0912	Inverse Agonist	Gavage, once daily (3, 10 mg/kg)	n = 24	19 days	Decrease	
Chen et al., 2017 [71]	DIO (male C57BL/6 J mice)	TXX-522	Inverse Agonist	Gavage, once daily (5, 10 mg/kg)	n = 32	4 weeks	Decrease	
Méndez-Díaz et al., 2015 [72]	Wild-type (male Wistar rats)	ENP11	Inverse Agonist	IP injection, once daily (0.5, 1, 3 mg/kg)	n = 40	Acute	Not assessed	
Zhang et al., 2018 [73]	DIO (mice)	Compound 6a	Inverse Agonist	Oral, once daily (30 mg/kg)	unknown	5 days	Decrease	
Aceto et al., 2001 [74]	Wild-type (male Sprague–Dawley rats)	WIN 55,212-2	Agonist	IP injection, once daily (1, 2, 4, 8, 16, mg/kg)	n = 82	4 days	Decrease	
Abalo et al., 2009 [75]	Wild-type (male Wistar rats)	WIN 55,212-2	Agonist	IP injection, once daily (0.5, 5 mg/kg)	n = 56	14 days	Decrease	
Abalo et al., 2013 [76]	Wild-type (male Wistar rats)	WIN 55,212-2	Agonist	IP injection, once weekly (0.5, 1 mg/kg)	n = 54	4 weeks	Decrease	Intensified weight loss from cisplatin
Radziszewska et al., 2014 [77]	Wild-type (male Wistar rats)	WIN 55,212-2	Agonist	IP injection, once daily (1 mg/kg)	n ~ 32	3 days	Decrease	
		AM 251	Inverse Agonist	IP injection, once daily (1 mg/kg)			Decrease	
Radziszewska et al., 2013 [78]	Wild-type (male Wistar rats)	WIN 55,212-2	Agonist	IP injection, once daily (0.5, 1, 2, 4 mg/kg)	unknown	Acute	Decrease	
Segev et al., 2014 [79]	Chronic mild stress (male Sprague–Dawley rats)	WIN 55,212-2	Agonist	IP injection, once daily (0.5 mg/kg)	unknown	3 days	No change	WIN 55,212-2 and AM 251 were coadministered
		AM 251	Inverse Agonist	IP injection, once daily (0.3 mg/kg)				
Jahanabadi et al., 2016 [80]	Diabetes (male Wistar albino rats)	WIN 55,212-2	Agonist	Intrathecal injection (1, 10, 100 µg/10 µL)	n ~ 28	Acute	No change	
Argueta et al., 2019 [81]	Wild-type (C57BL/6 J mice)	WIN 55,212-2	Agonist	IP injection, once daily (3 mg/kg)	n ~ 20	60 days	Not assessed	Study of satiation peptide response
		AM6545	Neutral Antagonist	IP injection, once daily (10 mg/kg)				
de Sousa Cavalcante et al., 2020 [82]	Cachexia (male Wistar rats)	WIN 55,212-2	Agonist	Subcutaneous injection, once daily (2 mg/kg)	n ~ 64	1 week	Decrease	No change in body weight in cachexia induced rats
Dalton et al., 2009 [83]	Wild-type (male Wistar rats)	HU-210	Agonist	IP injection, once daily (25, 50, 100 µg/kg)	n = 40	2 weeks	Decrease	
Giuliani et al., 2000 [84]	Wild-type (male Wistar rats)	HU-210	Agonist	IP injection, once daily (25, 50, 100 µg/kg)	n = 32	4 days	Decrease	
del Arco et al., 2000 [85]	Pregnancy (female Wistar rats)	HU-210	Agonist	IP injection, once daily (1, 5, 25 µg/kg)	unknown	>70 days	Decrease	
Scherma et al., 2017 [86]	Activity-based anorexia (female Sprague–Dawley rats)	CP-55,940	Agonist	IP injection, once daily (0.03, 0.06 mg/kg)	n = 168	6 days	Increase	Both caused decrease in body weight loss compared to vehicle
		THC	Partial Agonist	IP injection, once daily (0.5, 0.75 mg/kg)			Increase	
Takeda et al., 2015 [87]	Wild-type and Apo-E deficient (male BALB/c mice)	CBDD	Agonist	Oral, once daily (0.025, 0.25 mg/kg)	n = 12	~24 weeks	Increase	
Järbe et al., 2005 [24]	Wild-type (male Sprague–Dawley rats)	THC	Partial Agonist	IP injection, once daily (0.1-1.8 mg/kg)	n = 32	6 days	Decrease	THC and rimonabant administered separately and together
		Rimonabant	Inverse Agonist	IP injection, once daily (0.03-0.3 mg/kg)				
Lewis et al., 2010 [88]	Activity-based anorexia (male C57BL/6 J mice)	THC	Partial Agonist	IP injection, once daily (0.5 mg/kg)	n = 32	8 days	Increase	
Verty et al., 2011 [89]	Activity-based anorexia (female Sprague–Dawley rats)	THC	Partial Agonist	IP injection, once daily (0.1, 0.5, 2 mg/kg)	n = 28	6 days	Increase	
Wong et al., 2012 [90]	Wild-type (Australian Albino Wistar rats)	THC	Partial Agonist	IP injection, once daily (10 mg/kg)	n = 10	10 days	Decrease	
Coskun and Bolkent, 2014 [91]	Diabetes (rats)	THC	Partial Agonist	IP injection, once daily (3 mg/kg)	n = 29	7 days	Increase	
Keeley et al., 2015 [92]	Puberty (male and female Long–Evans and Wistar rats)	THC	Partial Agonist	IP injection, once daily (5 mg/kg)	n = 335	2 weeks	Decrease	
Cluny et al., 2015 [93]	DIO and lean (male C57BL/6N mice)	THC	Partial Agonist	IP injection, once daily (2 mg/kg for 3 weeks, 4 mg/kg for 1 week)	n = 32	4 weeks	Decrease in DIO mice	No effect on body weight in lean mice
Marcus et al., 2016 [94]	Hyper-sensitive CB1 (male S426A/S430A mice)	THC	Partial Agonist	IP injection (1, 3, 10 mg/kg)	unknown	Acute	No change	
Beydogan et al., 2019 [95]	High-fructose diet (male Sprague–Dawley rats)	THC	Partial Agonist	IP injection, once daily (1.5 mg/kg)	n = 32	12 weeks (THC administration for final 4 weeks)	Decrease	
Nguyen et al., 2020 [96]	Adolescence (female and male Wistar rats)	THC	Partial Agonist	Vapour inhalation (30 mins, twice daily, 5 days/week)	n = 88	2 weeks	Decrease in males	
Ogden et al., 2019 [97]	Wild-type (female Long–Evans rats)	AM11101	Partial Agonist	IP injection, (0.1 mg/kg)	n = 21	1 week	No change	
		THC	Partial Agonist	IP injection, (1 mg/kg)			No change	
Pavon et al., 2006 [98]	DIO (Zucker rats) and Wild-type (male Wistar rats)	LH-21	Neutral Antagonist	IP injection, once daily (0.03, 0.3, 3 mg/kg)	unknown	8 days	Decrease in DIO rats	
Alonso et al., 2012 [99]	DIO (male Wistar rats)	LH-21	Neutral Antagonist	IP injection, once daily (3 mg/kg)	n = 32	10 days	Decrease	
Chen et al., 2008 [100]	Wild-type (C57BL/6 J mice)	LH-21	Neutral Antagonist	IP injection (10, 30, 60 mg/kg)	n = 45	Acute	Decrease	
Romero-Zerbo et al., 2017 [101]	DIO, pre-diabetes (C57BL/6 J mice)	LH-21	Neutral Antagonist	IP injection (3 mg/kg)	n ~ 30	2 weeks	No change	
Dong et al., 2018 [102]	DIO, hypertension (female C57BL/6 J mice)	LH-21	Neutral Antagonist	IP injection (1, 3 mg/kg)	n = 8	3 weeks	Decrease	
Cluny et al., 2010 [103]	Wild-type (male Sprague–Dawley rats)	AM6545	Neutral Antagonist	IP injection (10 mg/kg)	n = 8	1 week	Decrease	
Tam et al., 2010 [104]	DIO (male C57BL/6 J mice)	AM6545	Neutral Antagonist	IP injection (10 mg/kg)	n = 40	4 weeks	Decrease	
		Rimonabant	Inverse Agonist	IP injection (10 mg/kg)			Decrease	
Ma et al., 2018 [105]	DIO (ICR mice)	AM6545	Neutral Antagonist	IP injection (3, 10 mg/kg)	n = 32	3 weeks	Decrease	
Chambers et al., 2007 [106]	Wild-type (male Sprague–Dawley rats)	AM4113	Neutral Antagonist	IP injection (1, 5, 10, 20 mg/kg)	n = 39	5 days	Decrease	
		AM 251	Inverse Agonist	IP injection (5 mg/kg)			Decrease	
Sink et al., 2008 [107]	Wild-type (male Sprague–Dawley rats)	AM4113	Neutral Antagonist	IP injection (2, 4, 8 mg/kg)	n = 30	Acute	Not assessed	
Cluny et al., 2011 [108]	Wild-type (male Sprague–Dawley rats)	AM4113	Neutral Antagonist	IP injection (2, 10 mg/kg)	n = 17	2 weeks	Decrease	
Gueye et al., 2016 [109]	Nicotine Dependence (male Long–Evans and Wistar rats)	AM4113	Neutral Antagonist	IP injection (1, 3, 10 mg/kg)	n = 149	3 weeks	Decrease	
		Rimonabant	Inverse Agonist	IP injection (1, 3, 10 mg/kg)			Decrease	
Balla et al., 2018 [110]	Alcoholism (male C57BL/6 J mice)	AM4113	Neutral Antagonist	IP injection (1, 3 mg/kg)	n = 31	4 days	No change	
Wargent et al., 2013 [111]	DIO (female C5BL/6 J mice)	THCV	Neutral Antagonist	Gavage, once or twice daily (0.1–12.5 mg/kg)	n = 63	45 days	No change	
		AM 251	Inverse Agonist	Gavage, twice daily (10 mg/kg)		45 days	Decrease	
Mastinu et al., 2013 [112]	DIO (male C57BL/6 N mice)	NESS06SM	Neutral Antagonist	Gavage, once daily (10, 30 mg/kg)	n = 60	30 days	Decrease	
		Rimonabant	Inverse Agonist	Gavage, once daily (10 mg/kg)			Decrease	
Fois et al., 2016 [113]	Wild-type (male Sprague–Dawley rats)	SM-11	Neutral Antagonist	IP injection (0.05, 0.125, 0.25 mg/kg)	n = 32	10 days	Decrease	
Seltzman et al., 2017 [114]	DIO (male C57BL/6 J mice)	PIMSR	Neutral Antagonist	IP injection (10 mg/kg)	n = 12	4 weeks	Decrease	

**Table 2 biomolecules-10-00855-t002:** Summary of Evidence: Clinical Studies of Cannabinoid Drug Effect on Body Weight.

Study Reference	Study Design	Population Characteristics	Cannabinoid Administered	Cannabinoid Type	Drug Administration	Duration	Effect on Body Weight, Compared to Placebo (If Applicable)	Other Notes
Greenberg et al., 1976 [115]	Observational	Healthy males (n = 37)	Cannabis (1.8–2.3% THC)	Cannabis	Ad libitum	21 days	Increase	
Foltin et al., 1986 [21]	Double-blinded, Placebo-controlled	Healthy males (n = 9)	Cannabis (1.84% THC)	Cannabis	Uniform puff procedure	25 days	No change	
Foltin et al., 1988 [116]	Double-blinded, Placebo-controlled	Healthy males (n = 6)	Cannabis (2.3% THC)	Cannabis	Uniform puff procedure	13 days	Increase	
Le Strat and Le Foll, 2011 [27]	Cross-Sectional	Population Representative (n = 50,736)	Cannabis	Cannabis	N/A	N/A	Decrease	
Warren et al., 2005 [117]	Retrospective Chart Review	Females referred for weight management (n = 297)	Cannabis	Cannabis	N/A	N/A	Decrease	
Rodondi et al., 2006 [118]	Longitudinal	Black and White Adults 18–30 (n = 3617)	Cannabis	Cannabis	N/A	15 years	Decrease	Study of coronary artery disease risk factors
Penner et al., 2013 [119]	Cross-sectional	Population Representative (n = 4657)	Cannabis	Cannabis	N/A	N/A	Decrease	
Hayatbakhsh et al., 2010 [28]	Prospective Cohort	Young adults (n = 2566)	Cannabis	Cannabis	N/A	21 years	Decrease	Followed from birth to 21 years
Huang et al., 2013 [120]	Longitudinal	Adolescents (n = 5141)	Cannabis	Cannabis	N/A	12 years	Increase	Increased trajectory of adolescent cannabis use associated with obesity
Muniyappa et al., 2013 [121]	Cross-sectional, case-control	BMI-matched cannabis smokers and non-smokers (n = 60)	Cannabis	Cannabis	N/A	N/A	No change	Greater abdominal visceral fat in cannabis smokers
Cobb et al., 2019 [122]	Survey	African American > 55 years (n = 340)	Cannabis	Cannabis	N/A	N/A	Decrease	
Racine et al., 2015 [30]	Cross-Sectional	African American adults (n = 100)	Cannabis	Cannabis	N/A	N/A	No change	Insignificant trend towards lower BMI in current cannabis users
Ngueta et al., 2015 [123]	Cross-Sectional	Inuit adults (n = 786)	Cannabis	Cannabis	N/A	N/A	Decrease	
Ross et al., 2017 [124]	Longitudinal	Adult cannabis users (n = 238)	Cannabis	Cannabis	N/A	2 years	Increase	
Alshaarawy and Anthony, 2019 [125]	Longitudinal	Population Representative (n = 33,000)	Cannabis	Cannabis	N/A	3 years	Decrease	Longitudinal study of NESARC and NCS-R
Meier et al., 2019 [126]	Longitudinal	Young males (n = 253)	Cannabis	Cannabis	N/A	25 years	Decrease	
Bancks et al., 2018 [127]	Longitudinal	Healthy adults 18–30 (n = 2902)	Cannabis	Cannabis	N/A	25 years	Decrease	
Thompson and Hay, 2015 [128]	Cross-Sectional	Population Representative (n = 6281)	Cannabis	Cannabis	N/A	7 years	Decrease	
N’Goran et al., 2015 [129]	Longitudinal	Young males (n = 7563)	Cannabis	Cannabis	N/A	15 months	N/A	Greater BMI increased chances of increased cannabis use
Jin et al., 2017 [130]	Longitudinal	Young males (n = 712)	Cannabis	Cannabis	N/A	20–22 years	No change	
Vázquez-Bourgon et al., 2019 [131]	Longitudinal	First-episode non-affective psychosis patients (n = 510)	Cannabis	Cannabis	N/A	3 years	Decrease	All subjects treated with oral antipsychotic medication
Vázquez-Bourgon et al., 2019 [132]	Longitudinal	First-episode non-affective psychosis patients (n = 390)	Cannabis	Cannabis	N/A	3 years	Decrease	Follow-up study evaluating non-alcoholic fatty liver disease
Scheffler et al., 2018 [133]	Longitudinal	Antipsychotic-naïve psychiatric patients (n = 109)	Cannabis	Cannabis	N/A	1 year	Decrease	
Bruins et al., 2016 [134]	Longitudinal	Adults with severe mental illness (n = 3169)	Cannabis	Cannabis	N/A	~14 months	Decrease	
Kindred, 2017 [135]	Survey	Parkinson’s and multiple sclerosis patients (n = 595)	Cannabis	Cannabis	N/A	N/A	Decrease	
Ngueta and Ndjaboue, 2019 [136]	Cross-Sectional	Population Representative (n = 129,509)	Cannabis	Cannabis	N/A	N/A	Decrease	
Danielsson et al., 2016 [137]	Longitudinal	Healthy adults 18–84 (n = 17,967)	Cannabis	Cannabis	N/A	8 years	Decrease	
Van Gaal et al., 2005 [138]	Double-blinded, Placebo-controlled, multicentre	Adults BMI ≥ 30 or ≥ 27 kg/m2 with comorbidity (n = 920)	Rimonabant	Inverse Agonist	Oral (5, 20 mg/day)	1 year	Decrease	
Van Gaal et al., 2008 [139]	Double-blinded, Placebo-controlled, multicentre	Adults BMI ≥ 30 or ≥ 27 kg/m2 with comorbidity (n = 1508)	Rimonabant	Inverse Agonist	Oral (5, 20 mg/day)	2 years	Decrease	
Pi-Sunyer et al., 2006 [140]	Double-blinded, Placebo-controlled, multicentre	Adults BMI ≥ 30 or ≥ 27 kg/m2 with comorbidity (n = 3045)	Rimonabant	Inverse Agonist	Oral (5, 20 mg/day)	2 years	Decrease	
Van Gaal et al., 2008 [141]	Double-blinded, Placebo-controlled, multicentre	Adults BMI ≥ 30 or ≥ 27 kg/m2 with comorbidity (n = 6627)	Rimonabant	Inverse Agonist	Oral (5, 20 mg/day)	2 years	Decrease	Pooled from all RIO studies
Bergholm et al., 2013 [142]	Double-blinded, Placebo-controlled	Obese adults (n = 37)	Rimonabant	Inverse Agonist	Oral (20 mg/day)	48 weeks	Decrease	
Topol et al., 2010 [143]	Double-blinded, Placebo-controlled, multicentre	Obese adults (n = 18,695)	Rimonabant	Inverse Agonist	Oral (20 mg/day)	13.8 months (mean follow-up)	Not assessed	Discontinued due to adverse psychiatric side effects
Heppenstall et al., 2012 [144]	Open label	Obese adults with type 2 diabetes (n = 20)	Rimonabant	Inverse Agonist	Oral (20 mg/day)	6 months	Decrease	
Hollander et al., 2010 [145]	Double-blinded, Placebo-controlled, multicentre	Type 2 diabetic adults (n = 368)	Rimonabant	Inverse Agonist	Oral (20 mg/day)	48 weeks	Decrease	
Scheen et al., 2006 [146]	Double-blinded, Placebo-controlled, multicentre	Type 2 diabetic adults (n = 692)	Rimonabant	Inverse Agonist	Oral (5, 20 mg/day)	1 year	Decrease	
Proietto et al., 2010 [147]	Double-blinded, Placebo-controlled, multicentre	Obese adults (n = 693)	Taranabant	Inverse Agonist	Oral (0.5, 1, 2 mg/day)	1 year	Decrease	
Aronne et al., 2010 [148]	Double-blinded, Placebo-controlled, multicentre	Obese adults (n = 2502)	Taranabant	Inverse Agonist	Oral (2, 4, 6 mg/day)	2 years	Decrease	Weight loss did not increase significantly during second year of treatment
Wadden et al., 2010 [149]	Double-blinded, Placebo-controlled, multicentre	Obese adults (n = 784)	Taranabant	Inverse Agonist	Oral (0.5, 1, 2 mg/day)	1 year	Decrease	
Addy, Wright et al., 2008 [150]	Double-blinded, Placebo-controlled	Healthy male adults (n = 15)	Taranabant	Inverse Agonist	Oral (0.5, 2, 4, 6, 7.5 mg/day)	12 weeks	Decrease	
Addy, Li et al., 2008 [151]	Double-blinded, Placebo-controlled	Healthy male adults (n = 24)	Taranabant	Inverse Agonist	Oral (0.5–600 mg)	Acute	No change	
Addy, Rothenberg et al., 2008 [152]	Double-blinded, Placebo-controlled	Healthy male adults (n = 60)	Taranabant	Inverse Agonist	Oral (5, 7.5, 10, 25 mg/day)	2 weeks	Not assessed	
Kipnes et al., 2010 [153]	Double-blinded, Placebo-controlled, multicentre	Obese adults with type 2 diabetes (n = 623)	Taranabant	Inverse Agonist	Oral (0.5, 1, 2 mg/day)	1 year	Decrease	
Klumpers et al., 2013 [154]	Double-blinded, Double Dummy, Placebo-controlled	Healthy male cannabis users (n = 24)	TM38837	Inverse Agonist	Oral (100, 500 mg)	Acute	Not assessed	
Bedi et al., 2010 [22]	Double-blinded, Within-subject	HIV-positive cannabis users (n = 7)	Dronabinol	Partial Agonist	Oral (20 mg/day 2 days, 40 mg/day 14 days)	16 days	No change	
Haney et al., 2005 [23]	Double-blinded, Within-subject	HIV-positive cannabis users (n = 30)	Dronabinol	Partial Agonist	Oral (10, 20, 30 mg/day)	3–4 weeks	Not assessed	
Cannabis	Cannabis	Smoked (1.8, 2.8, 3.9% THC)
DeJesus et al., 2007 [155]	Retrospective Chart Review	HIV-positive subjects (n = 155)	Dronabinol	Partial Agonist	Oral (9.6–10.8 mg/day)	12 months	Increase	
Haney et al., 2007 [156]	Double-blinded, Within-subject	HIV-positive cannabis users (n = 10)	Dronabinol	Partial Agonist	Oral (5, 10 mg/day)	6 weeks	Increase	
Cannabis	Cannabis	Smoked (2.0, 3.9% THC)	Increase	
Andries et al., 2014 [157]	Double-blinded, Placebo-controlled, crossover	Anorexic women (n = 25)	Dronabinol	Partial Agonist	Oral (5 mg/day)	12 weeks	Increase	
Reichenbach et al., 2015 [158]	Double-blinded, Placebo-controlled	Noncardiac chest pain subjects (n = 13)	Dronabinol	Partial Agonist	Oral (5 mg/day)	4 weeks	No change	
Howard et al., 2019 [159]	Retrospective, Observational	Suppressed appetite patients (n = 38)	Dronabinol	Partial Agonist	Oral (mean 2.91 mg/day)	9.5 days (mean)	No change	
Côté et al., 2016 [160]	Double-blinded, Placebo-controlled	Chemotherapy patients (n = 65)	Nabilone	Partial Agonist	Oral (0.5–2 mg/day)	11 weeks	No change	
Levin et al., 2017 [161]	Double-blinded, Placebo-controlled	Postoperative nausea and vomiting patients (n = 340)	Nabilone	Partial Agonist	Oral (0.5 mg)	Acute	Not assessed	
Rzepa et al., 2015 [162]	Double-blinded, Placebo-controlled, Within-subject	Healthy adults 20–36 (n = 19)	THCV	Neutral Antagonist	Oral (10 mg)	Acute	Not assessed

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
