# Peer review of "Targeting the Endocannabinoid CB1 Receptor to Treat Body Weight Disorders: A Preclinical and Clinical Review of the Therapeutic Potential of Past and Present CB1 Drugs"

_biomolecules, 2020, doi:10.3390/biom10060855_

Round 1
Reviewer 1 Report
The manuscript entitled “Targeting the Endocannabinoid System to Treat Obesity: a Preclinical and Clinical Review of the Impact of Cannabinoid CB1 Drugs on Body Weight” is a comprehensive review of the current knowledge of cannabinoids and their role in obesity. Overall the manuscript is well-written and organized. A strength of the review is the careful consideration of both animal and human studies. A few comments should be addressed.
- There is no reference to Table 1 in the manuscript.
- The use of “psychiatric” to describe the adverse CNS effects of cannabinoids seemed ill-defined in its first few uses. However, it is nicely defined in lines 216-218. Could this description be moved closer to the front of the text, for instance in line 90?
- In line 57 in which CB2 is described in the immune system, it is stated that CB2 are classically considered to be located in T cells. It is true that CB2 is expressed in T cells but of all the immune cells, T cells are among the lowest expressors of CB2. It would be better to change this statement to “….cells of the immune system,…”. In other words delete “T”.
- In line 86 suggest replacing “famous” with “well-known” or “best characterized”.
- In line 248 “stimulation” should be replaced with “treatment” and in line 250 it should be ”concentration” not “dose” since these experiments were conducted in culture.
- The paragraph starting on line 268 should be broken up into two paragraphs since it starts with inflammation but then switches to discuss the CNS. Otherwise these two ideas need to be linked better.
- Line 322 “ARPEGGIO” should be defined.
- In line 488 the “cachexia index” should be defined so the reader has a better understanding of the endpoint.
- In section 3.2.2. there are some data that are inconsistent. Perhaps reorganization of this paragraph with a few transition statements to provide context are needed. For instance the first study discusses weight loss, then the CP study was conducted to treat anorexia, then the last study with CBDD produced weight gain. It is difficult to follow just stated as a list of studies.
- The THC section is not organized by pre-clinical and clinical.
- In lines 624 in which the effects of AM11101 are described the species is not indicated.
- In section 3.4.3. it is not always clear how some of the described studies (especially in lines 686-700) relate to obesity. Could the authors provide some context or relate the studies to obesity?
Author Response
Thank you for your revisions. Please see the attachment

Reviewer 2 Report
The MS deals with one of the most controversial topic of cannabinoids research, namely their effect on body weight. The literature on this topic is difficult to read and controversial, and the MS manages very well to summarize it. I have summarized a series of issues that the authors should consider for the preparation of a revised version
- The mechanistic details of the paradoxical anti-obesity effects of CB1 agonists are unknown, but some hypotheses have been put forward, and are worth mentioning
- THCAA should be included in the agonistic cannabinoids active in weight management, since it showed excellent activity in animal model studies
- Limitations of observational studies related to the different ways recreational cannabis is consumed in Europe (diluted with tobacco) and US (in purity) should also be considered in the rationalization of the discrepancies between marijuana studies.
- There is, in general, little coverage of the molecular details involved in the regulation of body weight by cannabinoids and the cannabinoid system, despite the various hints from the rimonabant studies, including those on brown fat.
- the conclusions are realistic and well articulated. Nevertheless, since the authors have played a preeminent role in the field, they could also outline futures methodological issues for clarifying the complex topic of their review. Overall, a very good review on a difficult and controversial topic, that I am pleased to recommend for publication with only minor changes
Author Response
Thank you for your revisions. Please see the attachment.

Reviewer 3 Report
In the article “Targeting the Endocannabinoid System to Treat Obesity: a Preclinical and Clinical Review of the Impact of Cannabinoid CB1 Drugs on Body Weight” Murphy and Le Foll review the numerous preclinical and clinical studies on the effect of different CB1 ligands on body weight regulation, including the best studied inverse agonists, and also neutral antagonists, agonists and partial agonists. While description of the studies with different drugs is accurate and complete, critical discussion is limited, and strategies for future research are mostly lacking. Besides these general observations, I have some specific recommendations and requests:
- No Figures are present: a scheme with the major milestones through the past and present studies performed with CB1 drugs for the treatment of obesity could enhance understanding
- Since the Authors describe the current findings on the effects of CB1 agonists and antagonists, showing both anorexigenic and orexigenic effects, the title could be better modified in:
“Targeting the cannabinoid CB1 receptor to treat body weight disorders: a preclinical and clinical review of the therapeutic potential of past and present CB1 drugs”
- After discussing the problem of Rimonabant side effects, please clarify better that interest has now moved toward peripherally restricted CB1 blockers
- Page 2, lane 53: two “main” cannabinoid receptors…, “two major” endogenous lipid-based… Please remove “retrograde neurotransmitters” since endocannabinoids in peripheral tissues act in autocrine or paracrine mode.
- Page 2, lane 55: enzymes instead of processes
- Page 2, lane 57: add “adipose tissue” to the list before ref [8]
- Page 2, lane 57: CB2 receptors “were” instead of “are”
- Page 2, lane 63: add that AEA is also a full agonist for vanilloid receptors
- Page 7, lane 211: add a brief explanation of what inverse agonists are
- Page 13, lane 413: the title “Promising CB1 inverse agonists of recent years” should include the concept of “peripherally restricted CB1 blockers”
- Page 14, lanes 456-458: move before 3.2.1.
- Page 15: delete lanes 493-494 (already said)
- Page 16, lane 514: After “CB1 partial agonist” add Ref and “Notably, THC binds also to many other receptors (GPR1B, PPARg, TRPA, TRPV, etc.)” and add Ref (this information can be useful to better understand some complex THC effects)
- Page 19, lanes 628-631: move before 3.4.1.
Author Response

(The authors gave the same response as above.)
